# Adaptive Proximal Gradient Methods for Structured Neural Networks

**Jihun Yun**
KAIST
arcprime@kaist.ac.kr

**Aurélie C. Lozano**
IBM T.J. Watson Research Center
aclozano@us.ibm.com

**Eunho Yang**
KAIST, AITRICS
eunhoy@kaist.ac.kr

## Abstract

We consider the training of structured neural networks where the regularizer can be non-smooth and possibly non-convex. While popular machine learning libraries have resorted to stochastic (adaptive) subgradient approaches, the use of proximal gradient methods in the stochastic setting has been little explored and warrants further study, in particular regarding the incorporation of adaptivity. Towards this goal, we present a general framework of stochastic proximal gradient descent methods that allows for arbitrary positive preconditioners and lower semi-continuous regularizers. We derive two important instances of our framework: (i) the first proximal version of ADAM, one of the most popular adaptive SGD algorithm, and (ii) a revised version of PROXQUANT [1] for quantization-specific regularizers, which improves upon the original approach by incorporating the effect of preconditioners in the proximal mapping computations. We provide convergence guarantees for our framework and show that adaptive gradient methods can have faster convergence in terms of constant than vanilla SGD for sparse data. Lastly, we demonstrate the superiority of stochastic proximal methods compared to subgradient-based approaches via extensive experiments. Interestingly, our results indicate that the benefit of proximal approaches over sub-gradient counterparts is more pronounced for non-convex regularizers than for convex ones.

## 1 Introduction

We study the regularized training of neural networks, which can be formulated as the following (stochastic) optimization problem

$$\underset{\theta \in \mathbb{R}^d}{\text{minimize}} \ F(\theta) \coloneqq \overbrace{\mathbb{E}_{\xi \sim \mathbb{P}}\big[f(\theta; \xi)\big]}^{f(\theta)} + \mathcal{R}(\theta) \tag{1}$$

where $\theta \in \mathbb{R}^d$ represents the network parameter, $\xi$ is the random variable representing mini-batch data samples, and $\mathcal{R}(\cdot)$ is a regularizer encouraging low-dimensional structural constraints on $\theta$.

The technique of regularization is ubiquitous in machine learning as it can effectively prevent overfitting and yield better generalization. The $\ell_1$-regularized training for Lasso estimators/sparse Gaussian graphical model (GMRF) estimation [2, 3] and $\ell_2$ weight decay [4] on parameters are prototypical examples. In the context of deep learning, important instances include network pruning [5, 6], which induces a sparse network structure, and network quantization [7, 8, 1], which gives hard constraints so that parameters have only discrete values.

For the *unregularized* case, i.e., when $\mathcal{R}(\theta) = 0$, stochastic gradient descent (SGD) has been a prevalent approach to solve the optimization problem stated in (1). At each iteration, SGD evaluates the gradient on a randomly chosen subset of training samples (mini-batch). While vanilla SGD employs a uniform learning rate for all coordinates, several adaptive variants have been proposed to

Table 1: Comparison among *stochastic* (or *online*) PGD for solving the problem in (1).

| Algorithm | Non-convex Loss | Non-convex Regularizer | Arbitrary Preconditioner | Momentum | Convergence Guarantee |
|---|---|---|---|---|---|
| ADAGRAD [9] | ✗ | ✗ | △ (ADAGRAD) | ✗ | ✓ |
| [10] | ✓ | ✗ | ✓ | ✗ | ✓ |
| [11] | ✓ | ✗ | ✗ | ✓ | ✓ |
| [12] | ✓ | ✗ | ✗ | ✓ | ✓ |
| [13] | ✓ | ✗ | ✗ | ✓ | ✓ |
| [14] | ✓ | ✓ | ✗ | ✗ | ✓ |
| [15] | ✓ | ✓ | △ (ADAGRAD) | ✗ | ✓ |
| Prox-SGD [16] | ✓ | ✗ | ✓ | ✓ | ✗ |
| [17] | ✓ | ✓ | ✗ | ✓ | ✓ |
| PROXGEN (**Ours**) | ✓ | ✓ | ✓ | ✓ | ✓ |

dynamically take advantage of the data geometry by scaling the learning rate for each coordinate by its gradient history. Prime examples of such approaches include ADAGRAD [9], which adjusts the learning rate by the sum of all the past squared gradients, and exponential moving average (EMA) approaches such as RMSPROP [18] and ADAM [19], which scale down the gradients by square roots of exponential moving averages of squared past gradients to essentially limit the scope of the adaptation to only a few recent gradients. In terms of theory, convergence analyses of these unregularized SGD methods, whether adaptive or not, have been well studied both for convex [19, 20] and non-convex [21, 22] loss $f$ cases.

For the *regularized* case, since the regularizer is often *non-smooth* around some region (e.g. the $\ell_1$ norm), modern machine learning libraries such as TensorFlow [23] and PyTorch [24] therefore resort to using the *subgradient* of the objective function $F(\theta)$ in (1). Such a strategy is problematic as it may slow down convergence and result in oscillations.

A simple idea to bypass the non-smoothness of a regularizer is via its proximal operator. This idea is the basis of proximal gradient descent (PGD) methods, which first update the parameter using the gradient of the loss function $f(\theta)$ and then perform a proximal mapping of $\mathcal{R}(\theta)$. In the *non-stochastic* case, PGD with both convex and non-convex regularizers has been extensively studied in the literature [25, 26, 11, 12, 27]. Another work, VMFB [28], analyzes the preconditioned gradient descent on convex regularized problems with non-convex loss but does not consider the first-order momentum. In contrast, PGD in the *stochastic* setting has been little explored. [9, 10] consider PGD to solve the stochastic objectives with convex regularizers. Recently, [15] studies non-convex and non-smooth regularized problems for DC (difference of convex) functions and [14, 17] present non-asymptotic analysis for non-convex smooth loss and non-convex regularizers, which is the most general setting, but do not consider the preconditioner in the update rule.

All the aforementioned studies of the stochastic case, however, focus either on limited settings (e.g. [9] only covers the update rule of ADAGRAD) with convex regularizers only, or on pure vanilla gradient descent for non-convex regularizers. Hence, they cannot accommodate all advanced modern optimization algorithms with *preconditioners*, such as adaptive gradient methods. The only exception is PROX-SGD [16], with the caveat that PROX-SGD update rule is *not a pure* PGD. Moreover, the theory in [16] only guarantees convergence, *not how fast* Prox-SGD converges, and the analysis is performed *without* considering preconditioners.

In this paper, we propose an exact framework for stochastic proximal gradient methods with arbitrary positive preconditioners and lower semi-continuous (possibly non-convex) regularizers. With our framework, our goal is to provide theoretical and empirical understanding of stochastic proximal gradient methods for training structured neural networks. Our main contributions can be summarized as follows:

- We propose the first general family of stochastic proximal gradient methods, which we term PROXGEN. We introduce two important instances stemming from our approach: (i) the first proximal version of ADAM [19] and (ii) a revised version of PROXQUANT [1] that improves upon the original approach for quantization-specific regularizers by incorporating the effect of preconditioners when computing proximal mappings.

**Algorithm 1** PROXGEN: A **Gen**eral Stochastic **Prox**imal Gradient Method

---

1: **Input:** Stepsize $\alpha_t$, $\{\rho_t\}_{t=1}^{t=T} \in [0,1)$, regularization parameter $\lambda$, and small constant $0 < \delta \ll 1$.
2: **Initialize:** $\theta_1 \in \mathbb{R}^d$, $m_0 = 0 \in \mathbb{R}^d$, and $C_0 = O \in \mathbb{R}^{d \times d}$.
3: **for** $t = 1, 2, \ldots, T$ **do**
4:     Draw a minibatch sample $\xi_t$ from $\mathbb{P}$
5:     $g_t \leftarrow \nabla f(\theta_t; \xi_t)$                                                           ▷ Stochastic gradient
6:     $m_t \leftarrow \rho_t m_{t-1} + (1 - \rho_t) g_t$                            ▷ $1^{\text{st}}$-order momentum
7:     $C_t \leftarrow$ Preconditioner construction
8:     $\theta_{t+1} \in \underset{\theta \in \Omega}{\operatorname{argmin}} \left\{ \langle m_t, \theta \rangle + \lambda \mathcal{R}(\theta) + \frac{1}{2\alpha_t}(\theta - \theta_t)^{\mathsf{T}}(C_t + \delta I)(\theta - \theta_t) \right\}$     ▷ Update rule
9: **end for**

---

- We analyze the convergence of the general PROXGEN family and identify essential conditions for convergence. We show that in general PROXGEN enjoys the same convergence rate as vanilla SGD, but more importantly that the adaptive methods can have faster convergence in terms of constant than vanilla SGD for sparse data. Our convergence guarantee encompasses several existing approaches as special cases.

- In terms of practice, we demonstrate the superiority of proximal methods over subgradient-based methods with various non-convex regularizers which have not yet been studied in deep learning. Interestingly, our experiments indicate that the benefit of proximal methods over subgradient approaches is more pronounced with non-convex regularizers than with convex regularizers for learning sparse deep models.

Table 1 summarizes the previous studies and our work in terms of stochastic PGD.

## 2 A Unified Framework of Adaptive Proximal Gradient Methods

In this section, we present PROXGEN, a general family of stochastic proximal gradient methods, and present both existing and novel instances as showcase examples in our family. Algorithm 1 describes the details of PROXGEN. The update rule on line 8 of Algorithm 1 can be written more compactly:

$$\theta_{t+1} \in \underset{\theta \in \Omega}{\operatorname{argmin}} \left\{ \langle m_t, \theta \rangle + \lambda \mathcal{R}(\theta) + \frac{1}{2\alpha_t}(\theta - \theta_t)^{\mathsf{T}}\left(C_t + \delta I\right)(\theta - \theta_t) \right\}$$
$$= \operatorname{prox}_{\alpha_t \lambda \mathcal{R}(\cdot)}^{C_t + \delta I}\left(\theta_t - \alpha_t(C_t + \delta I)^{-1}m_t\right) \tag{2}$$

where the proximal operator in (2) is defined as $\operatorname{prox}_h^A(z) = \operatorname{argmin}_x\{h(x) + \frac{1}{2}\|x - z\|_A^2\}$. In PROXGEN, we allow both the loss and the regularizer to be non-convex. Now, we introduce possible examples according to the proper combinations of preconditioners $C_t$ and regularizers $\mathcal{R}(\cdot)$.

**Existing Instances of PROXGEN.** We briefly recover some known examples in PROXGEN family.

- ADAGRAD [9] is the first key instance of adaptive gradient methods where $C_t = (\sum_{\tau=1}^{t} g_\tau g_\tau^{\mathsf{T}})^{1/2}$ and $\mathcal{R}(\theta) = \|\theta\|_1$. Any convex regularizer $\mathcal{R}(\cdot)$ is allowed.

- The proximal Newton methods [29] employ the exact Hessian $C_t = \nabla^2 f(\theta_t)$ and $\mathcal{R}(\theta) = \|\theta\|_1$. In addition, we can approximate the exact Hessian, which yield proximal Newton-*type* methods such as quasi-Newton approximation [30], L-BFGS approximation [31], and adding a multiple of the identity to the Hessian.

Although the above examples enjoy good theoretical properties in convex settings, many of the modern practical optimization problems involve non-convex loss functions such as learning deep models. Moreover, it is known that non-convex regularizers yield better performance (also in terms of theory) than convex penalties in some applications (see [32, 33, 34, 35] and references therein). Considering this motivation and recent advanced optimizers, we arrive at the following new examples.

**Novel Instances of PROXGEN.** Beyond the well-known methods above, PROXGEN naturally introduces proximal versions of standard SGD techniques developed for solving unregularized problems for deep learning. The following examples are just a few instances that have not been

explored so far, and PROXGEN can cover a broader range of new examples depending on the combinations of preconditioners and regularizers.

- The *proximal version* of ADAM [19] with $\ell_q$ regularization is a possible example where $C_t = \sqrt{\beta C_{t-1} + (1-\beta)g_t^2}$ with $\beta \in [0,1)$ and $\mathcal{R}(\theta) = \|\theta\|_q$ for $0 \leq q \leq 1$. We validate empirically the superiority of our novel *proximal version* of ADAM over the usual subgradient-based counterpart in Section 4.

- We can also consider the *proximal* version of KFAC [36]. For an $L$-layer neural network, KFAC approximates the Fisher information matrix with layer-wise block diagonal structure where $l$-th diagonal block $C_{t,[l]}$ corresponds to Kronecker-factored approximation with respect to the parameters at $l$-th layer. The proximal version of KFAC corresponds to $C_{t,[l]} = \mathbb{E}[\boldsymbol{\delta}_l \boldsymbol{\delta}_l^\mathsf{T}] \otimes \mathbb{E}[\boldsymbol{a}_{l-1} \boldsymbol{a}_{l-1}^\mathsf{T}]$ and $\mathcal{R}(\theta) = \|\theta\|_q$ where $\boldsymbol{\delta}_l$ is the gradient with respect to the outputs of $l$-th layer and $\boldsymbol{a}_{l-1}$ is the activation of $(l-1)$-th layer.

**Examples of Proximal Mappings for PROXGEN.** We provide update rules for PROXGEN with $\ell_q$ regularization $(0 \leq q \leq 1)$ and diagonal preconditioners, for which closed-form updates are available. Diagonal preconditioners are used by popular adaptive gradient methods such as ADAM. Note, however, that our framework and convergence analysis are not limited to diagonal preconditioners and apply to general positive preconditioners. Specifically, we consider regularizer $\mathcal{R}(\theta) = \lambda \sum_{j=1}^p |\theta_j|^q$ for $\theta \in \mathbb{R}^p$ with diagonal preconditioner matrix $C_t$. Note that for $C_t = I$ (i.e. vanilla gradient descent), it is known that closed-form solutions exist for proximal mappings for $q \in \{0, \frac{1}{2}, \frac{2}{3}, 1\}$ [37]. We denote the $i$-th coordinate of the vector $\theta_t$ as $\theta_{t,i}$ and the diagonal entry $[C_t]_{ii}$ as $C_{t,i}$

- $\ell_1$ **regularization.** The proximal mappings for the case of $\ell_1$ regularization with preconditioner can be computed efficiently via soft-thresholding as

$$\widehat{\theta}_{t,i} = \theta_{t,i} - \alpha_t \frac{m_{t,i}}{C_{t,i} + \delta}, \quad \theta_{t+1,i} = \text{sign}(\widehat{\theta}_{t,i})\left(\left|\widehat{\theta}_{t,i}\right| - \frac{\alpha_t \lambda}{C_{t,i} + \delta}\right) \tag{3}$$

- $\ell_0$ **regularization.** In case of $\ell_0$ regularization, we can compute the closed-form solutions via hard-thresholding as

$$\widehat{\theta}_{t,i} = \theta_{t,i} - \alpha_t \frac{m_{t,i}}{C_{t,i} + \delta}, \quad \theta_{t+1,i} = \begin{cases} \widehat{\theta}_{t,i}, & |\widehat{\theta}_{t,i}| > \sqrt{\frac{2\alpha_t \lambda}{C_{t,i} + \delta}}, \\ 0, & |\widehat{\theta}_{t,i}| < \sqrt{\frac{2\alpha_t \lambda}{C_{t,i} + \delta}} \\ \{0, \widehat{\theta}_{t,i}\}, & |\widehat{\theta}_{t,i}| = \sqrt{\frac{2\alpha_t \lambda}{C_{t,i} + \delta}} \end{cases} \tag{4}$$

The closed-form proximal mappings for $\ell_{1/2}$ and $\ell_{2/3}$ regularization are provided in the Appendix.

**Revised PROXQUANT [1].** The recently proposed PROXQUANT proposes novel regularizations for network quantization. Especially for binary quantization, a W-shaped regularizer is defined as $\mathcal{R}_{\text{bin}}(\theta) = \|\theta - \text{sign}(\theta)\|_1$ where $\text{sign}(\theta)$ is applied on $\theta$ in an element-wise manner. Using this regularizer, the main difference between PROXQUANT and our PROXGEN approach is shown in Table 2.

Table 2: PROXQUANT versus *revised* PROXQUANT

| PROXQUANT | $\left\|\text{prox}_{\alpha_t \lambda \mathcal{R}(\cdot)}\left(\theta_t - \alpha_t(C_t + \delta I)^{-1}m_t\right)\right.$ |
|---|---|
| Revised PROXQUANT | $\left\|\text{prox}_{\alpha_t \lambda \mathcal{R}(\cdot)}^{C_t + \delta I}\left(\theta_t - \alpha_t(C_t + \delta I)^{-1}m_t\right)\right.$ |

Note that PROXQUANT (top in Table 2) does not consider the effect of preconditioners when computing proximal mappings. Therefore, we revise the proximal update in PROXQUANT by considering preconditioners in proximal mappings with PROXGEN (bottom in Table 2). Moreover, we also propose *generalized regularizers* motivated by $\ell_q$ regularization for $0 < q < 1$: $\mathcal{R}_{\text{bin}}^q(\theta) = \|\theta - \text{sign}(\theta)\|_q$. In terms of theory, [1] prove the convergence of PROXQUANT only for the *full-batch* gradient with *differentiable* regularizers, which is also guaranteed only for vanilla gradient descent. In contrast, using our *revised* PROXQUANT, we can completely bridge the gap in theory (via Theorem 1 in Section 3, which is stated for *stochastic* optimization), and we provide the *exact* update rule for solving the problem in (1). We also investigate the empirical differences of PROXQUANT and our revised PROXQUANT in Section 4.

# 3 Convergence Analysis

In this section, we provide convergence guarantees for the PROXGEN family. Our goal is to find an $\epsilon$-stationary point for the problem in (1) where $\epsilon$ is the required precision. For notational convenience, we assume that the regularization parameter $\lambda$ is incorporated into $\mathcal{R}(\theta)$ in (1). To guarantee the convergence under this setting, we should deal with the subdifferentials defined as:

**Definition 1** (Fréchet Subdifferential). *Let $\varphi$ be a real-valued function. The Fréchet subdifferential of $\varphi$ at $\bar{\theta}$ with $|\varphi(\bar{\theta})| < \infty$ is defined by*

$$\widehat{\partial}\varphi(\bar{\theta}) := \{\theta^* \in \Omega \mid \liminf_{\theta \to \bar{\theta}} \frac{\varphi(\theta) - \varphi(\bar{\theta}) - \langle \theta^*, \theta - \bar{\theta}\rangle}{\|\theta - \bar{\theta}\|} \geq 0\}.$$

**Definition 2** (Limiting Subdifferential). *Let $\widehat{\partial}\varphi(\bar{\theta})$ be the Fréchet subdifferential in Definition 1. The limiting subdifferential of $\varphi$ at $\bar{\theta}$ is defined by*

$$\partial\varphi(\bar{\theta}) := \{u \in \mathbb{R}^d : \exists \theta_k \xrightarrow{\varphi} \bar{\theta}, u_k \in \widehat{\partial}\varphi(\theta_k), u_k \to u\}.$$

*where $\theta_k \xrightarrow{\varphi} \bar{\theta}$ means $\theta_k \to \bar{\theta}$ with $\varphi(\theta_k) \to \varphi(\bar{\theta})$.*

To derive the convergence bound, we make the following mild conditions:

- **(C-1)** ($L$-smoothness) The loss function $f$ is differentiable, $L$-smooth, and lower-bounded: $\forall x, y, \|\nabla f(x) - \nabla f(y)\| \leq L\|x - y\|$ and $f(x^*) > -\infty$ for the optimal solution $x^*$.
- **(C-2)** (Bounded variance) The stochastic gradient $g_t = \nabla f(\theta_t; \xi)$ is unbiased and has the bounded variance: $\mathbb{E}_\xi[\nabla f(\theta_t; \xi)] = \nabla f(\theta_t)$, $\mathbb{E}_\xi[\|g_t - \nabla f(\theta_t)\|^2] \leq \sigma^2$.
- **(C-3)** (i) final step-vector is finite, (ii) the stochastic gradient is bounded, and (iii) the momentum parameter should be exponentially decaying: (i) $\|\theta_{t+1} - \theta_t\| \leq D$, (ii) $\|g_t\| \leq G$, (iii) $\rho_t = \rho_0 \mu^{t-1}$ with $D, G > 0$ and $\rho_0, \mu \in [0, 1)$.
- **(C-4)** (*Sufficiently positive-definite*) The minimum eigenvalue of effective spectrums should be uniformly lower bounded over all time $t$: $\forall t, \lambda_{\min}(\alpha_t(C_t + \delta I)^{-1}) \geq \gamma > 0$.

(C-1) and (C-2) are very standard in convergence analysis for optimization algorithms designed for deep learning such as ADAM, YOGI, and many others [21, 38, 39, 40, 41]. In addition, (C-3) is extensively studied in previous literature for analysis of general non-convex optimization [19, 20, 42, 40, 41]. Lastly, a similar condition to (C-4) is also considered in [39, 42]. We note that (C-3) and (C-4) are reasonable conditions: It is well-known that the parameter of an overparametrized neural network hardly changes from the initial point during training [43, 44, 45], so one can expect that the diameter $D$ of parameter space and the bound for the size of gradient $G$ have very small values and can be understood as *constants* in rates of the results. To validate this for real cases, we train ResNet-34 on CIFAR-10 dataset. In Figure-1-(a), the difference of parameters $\|\theta_{t+1} - \theta_t\|_2$ and the size of stochastic gradients $\|g_t\|_2$ attain just $1 \sim 3$ while the parameter dimension $d$ of ResNet-34 is about $10^7$. Hence, the constants $D$ and $G$ in (C-3) are negligible compared to the problem dimension $d$ in practice. The exponentially decaying momentum parameter assumption $\rho_t = \rho_0 \mu^{t-1}$ could be relaxed to $\rho_t = \rho_0/t$ sacrificing the logarithmic factor in our analysis. Also, (C-4) is indeed easily satisfied both theoretically and empirically. This condition holds in theory for most of the popular optimization algorithms for deep learning such as ADAGRAD, ADAM, and KFAC (the constant $\gamma$ is irrelevant to the problem dimension $d$ for each algorithm, and we defer the derivations to Appendix D). In order to investigate whether these conditions could be satisfied in real problems, we revisit the experiments of training ResNet-34. In Figure 1-(b), we can see the minimum eigenvalue of $\alpha_t(C_t + \delta I)^{-1}$ tends to increase, so the condition (C-4) is also satisfied empirically.

Since the loss function $f$ is assumed to be differentiable as in (C-1) and it is known that $\widehat{\partial}\varphi(\theta) \subseteq \partial\varphi(\theta)$, we have, at stationary points, $\mathbf{0} \in \widehat{\partial}F(\theta) = \nabla f(\theta) + \widehat{\partial}\mathcal{R}(\theta)$, so the convergence criterion is slightly different from that of general non-convex optimization. Hence, we use the following convergence criterion $\mathbb{E}[\text{dist}(\mathbf{0}, \widehat{\partial}F(\theta))] \leq \epsilon$ for an $\epsilon$-stationary point where $\text{dist}(x, A)$ denotes the distance between a vector $x$ and a set $A$. If no regularizer is considered ($\mathcal{R} = 0$), this criterion boils down to the one usually used in non-convex optimization, $\mathbb{E}[\|\nabla f(\theta)\|] \leq \epsilon$.

- **Challenges specific to the analysis of PROXGEN**. The most challenging issue in the analysis of PROXGEN compared to previous studies [14, 21] is that we should handle the momentum $m_t$ and

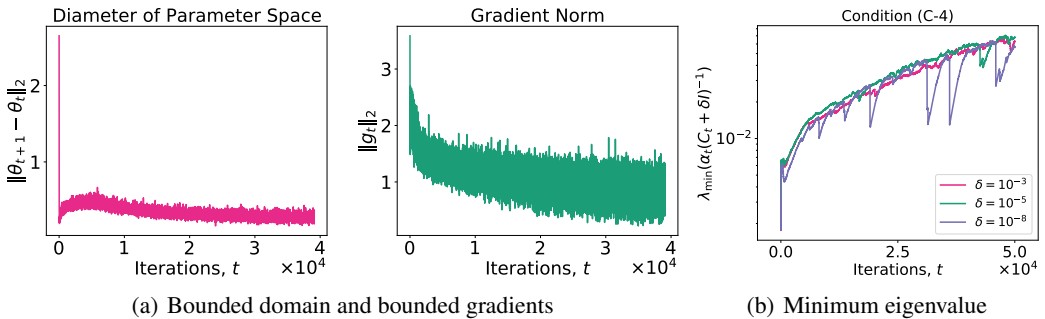

Figure 1: Empirical results for (a) condition (C-3) and (b) condition (C-4) using ResNet-34.

non-trivial preconditioner $C_t$. In terms of adaptive gradient methods, [21] guarantees the convergence of a family of adaptive methods (but without proximal mapping) using the changes of effective learning rate ($\Gamma_t := \alpha_t/\sqrt{V_t} - \alpha_{t+1}/\sqrt{V_{t+1}} \geq 0$ where $V_t$ is an adaptation matrix), which is a key quantity in their theory. [21] define a new sequence $\{z_t\}$ involving the quantity $\Gamma_t$ and exploit the simple closed-form of the quantity $z_{t+1} - z_t$ to derive the convergence with coordinate-wise analysis. However, this proof technique is not available to the *regularized* problems since $z_{t+1} - z_t$ is not amenable anymore to compute in a simple closed-form due to the proximal mapping. On the other hand, our proof *directly* solves the quadratic subproblem w.r.t. $\theta_t$ at line 8 in Algorithm 1 to handle a regularizer term. It should also be emphasized that our proof skill can handle arbitrary positive curvatures (hence including more general non-diagonal one) that were not acceptable in [21]. In the context of proximal gradient descent, our proof is totally different from [14] which is only for vanilla SGD. Due to the existence of $m_t$, it is highly non-trivial to bound the term $\|m_t - \nabla f(\theta_t)\|_2$ without suitable assumptions whereas $\|g_t - \nabla f(\theta_t)\|_2$ in [14] can be easily bounded using (C-2). Also, we need to deal with quadratic approximation term $(\theta - \theta_t)^\mathsf{T}(C_t + \delta I)(\theta - \theta_t)$ in Algorithm 1 which is not problematic in [14] simply because $C_t$ is trivially $I$. We could successfully bypass those difficulties using mild conditions (C-3) and (C-4), respectively.

We are ready to state our theorem for general convergence.

**Theorem 1.** *Let $\theta_a$ denote an iterate uniformly randomly chosen from $\{\theta_1, \cdots, \theta_T\}$. Under the conditions (C-1), (C-2), (C-3), (C-4) with the initial stepsize $\alpha_0 \leq \frac{\delta}{3L}$ and non-increasing stepsize $\alpha_t$, PROXGEN, Algorithm 1, is guaranteed to yield*

$$\mathbb{E}_a[\mathrm{dist}(\mathbf{0}, \widehat{\partial}F(\theta_a))^2] \leq \frac{Q_1\sigma^2}{T}\sum_{t=0}^{T-1}\frac{1}{b_t} + \frac{Q_2\Delta}{T} + \frac{Q_3}{T}$$

*where $\Delta = f(\theta) - f(\theta^*)$ with optimal point $\theta^*$, and $b_t$ is the minibatch size at time $t$. The constants $\{Q_i\}_{i=1}^3$ on the right-hand side depend on the constants $\{\alpha_0, \delta, L, D, G, \rho_0, \mu, \gamma\}$, but not on $T$.*

Note that the constants $\{Q_i\}_{i=1}^3$ in Theorem 1 are completely independent of the problem dimension $d$. From Theorem 1, the appropriate minibatch size is important to ensure a good convergence. Various settings for the minibatch size could be employed for convergence guarantee, but considering practical cases, we provide the following important corollary for *constant minibatch*.

**Corollary 1** (Constant Mini-batch). *Under the same assumptions as in Theorem 1 with sample size $n$ and constant minibatch size $b_t = b = \Theta(T)$, we have $\mathbb{E}_a\big[\mathrm{dist}(\mathbf{0}, \widehat{\partial}F(\theta_a))^2\big] \leq \mathcal{O}\big(1/T\big)$ and the total complexity is $\mathcal{O}(1/\epsilon^4)$ in order to have $\mathbb{E}_a\big[\mathrm{dist}\big(\mathbf{0}, \widehat{\partial}F(\theta_a)\big)\big] \leq \epsilon$.*

Here we make several remarks on our results and relationship with prior work.

• **On Convergence Results.** Note that our Corollary 1 achieves the optimal complexity $\mathcal{O}(1/\epsilon^4)$ of SGD to find $\epsilon$-stationary points under the standard assumptions (C-1) $\sim$ (C-4). Recent studies [46, 47] show faster rate, but under additional stronger assumptions such as second-order smoothness (i.e., the smoothness of Hessian matrix). Also, we could relax the exponentially decaying momentum $\rho_t = \rho_0\mu^{t-1}$ in (C-3) to $\rho_t = \rho_0/t$ as mentioned in [48] with the logarithm factor as $Q_3 = \mathcal{O}(\log T)$, which in result still ensures $\widetilde{\mathcal{O}}(1/\epsilon^4)$.

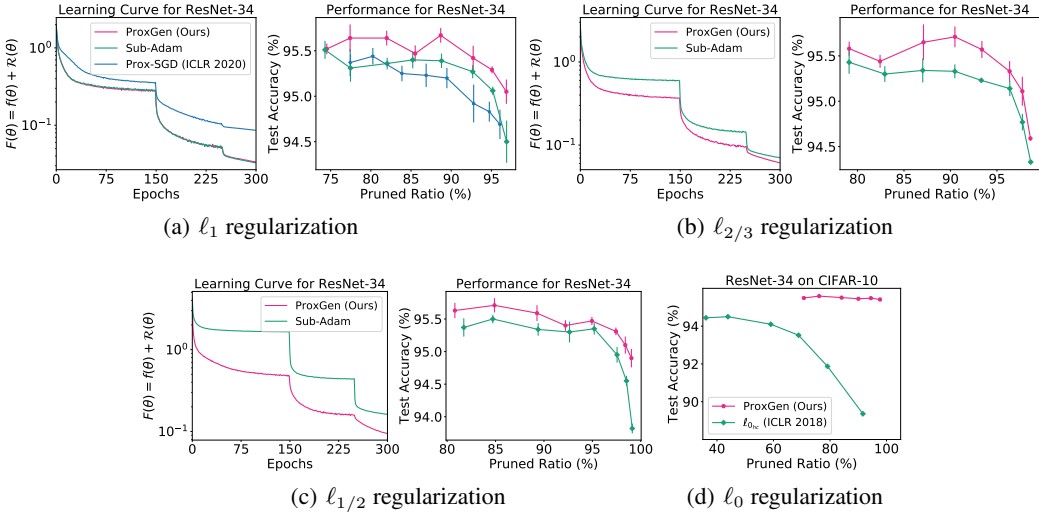

Figure 2: Comparison for sparse ResNet-34 on CIFAR-10 dataset with step-decay stepsize scheduling.

• **Advantages of using adaptive gradient methods in Theorem 1.** We discuss how the constant $\gamma$ in (C-4) affects the convergence in terms of theory. According to our proofs, $\gamma$ depends on the algorithmic details and the constants $Q_1, Q_2$ and $Q_3$ in Theorem 1 are proportional to $1/\gamma$. The convergence rate depends on these constants and the benefit of preconditioners can be found here. To view this more clearly, we consider the diagonal matrix adaptation of ADAM [19], i.e. constant stepsize $\alpha_t = \alpha$ and $C_t = \sqrt{(1-\beta)\sum_{\tau=1}^{t}\beta^{t-\tau}g_\tau \odot g_\tau}$, with $\beta \in [0, 1)$ and the total iteration $T$. In this setting, the $1/\gamma$ can be computed as

$$Q_i \propto \frac{1}{\gamma} = \frac{\sqrt{(1-\beta)\sum_{\tau=1}^{t}\beta^{t-\tau}\|g_\tau\|_2^2} + \delta}{\alpha} \leq \frac{G+\delta}{\alpha}$$

where $g_\tau$ is the gradient at time $\tau$ and $\delta$ is a small constant while the vanilla SGD ($C_t = 0$ and $\delta = 1$) satisfies $1/\gamma = 1/\alpha$. Here, we can clearly see the advantages of adaptive methods (i.e., using preconditioners) since $1/\gamma$ could be dramatically smaller if $\|g_\tau\|_2 \ll 1$ holds roughly with small constant $\delta$, which corresponds to sparse gradients $\|g_\tau\|_2$ (the data features are sparse). This coincides with the convex regret theory for adaptive gradient methods [9, 19, 48], which also holds in our theory with non-convex smooth loss and non-convex regularizers.

• **Implications of condition (C-4) on theory.** Our analysis relies on (C-4), the lower bound for the minimum eigenvalue of $\Gamma_t := \alpha_t(C_t + \delta I)^{-1}$. This means that Theorem 1 guarantees $\mathbb{E}_a[\text{dist}(\mathbf{0}, \widehat{\partial}F(\theta_a)^2] \leq \mathcal{O}(1/\sqrt{T})$ (in case of $b = \Theta(T)$ as in Corollary 1) for *any* change of basis of $\Gamma_t$, so in that sense, we provide a worst-case analysis and there is room for more optimistic bounds.

• **On minibatch in Corollary 1.** The conditions $b = \Theta(T)$ is considered as standard in many previous literature [38, 14] and is not stringent. In terms of stochastic optimization, it is natural in practice to choose the batch size $b$ and the number of epochs $e$ in advance. Then, the total number of iterations $T$ satisfies the following relation: $T = e \times \frac{n}{b} = e \times \frac{n}{\Theta(T)}$. In this sense, the total iterations $T$ should be an order of $\mathcal{O}(\sqrt{n})$ in practice. For example, this condition sets a minibatch size of approximately 200 and 1000 for CIFAR-10 and ImageNet dataset respectively, which is practical.

• **Connections to second-order methods.** Our analysis can provide guarantees for *positive* second-order preconditioners as long as (C-4) is satisfied (The empirical Fisher information [36] is one example). Although second-order solvers generally enjoy very fast convergence under strongly convex loss [29, 49], it can be understood that our theory guarantees *at least a sublinear rate for such second-order curvatures* with less stringent conditions.

Table 3: Comparison for binary neural networks. The best performance in mean value is highlighted.

| | | Test Error (%) | | | | |
| | | Baselines | | PROXGEN (Ours) | | |
| Model | Full Precision (32-bit) | BinaryConnect [8] | PROXQUANT [1] | Revised ProxQuant $\ell_1$ | Revised ProxQuant $\ell_{2/3}$ | Revised ProxQuant $\ell_{1/2}$ |
|---|---|---|---|---|---|---|
| ResNet-20 | 8.06 | $9.54 \pm 0.03$ | $\mathbf{9.35} \pm 0.13$ | $9.50 \pm 0.12$ | $9.72 \pm 0.06$ | $9.78 \pm 0.18$ |
| ResNet-32 | 7.25 | $8.61 \pm 0.27$ | $8.53 \pm 0.15$ | $8.29 \pm 0.07$ | $\mathbf{8.22} \pm 0.05$ | $8.43 \pm 0.15$ |
| ResNet-44 | 6.96 | $8.23 \pm 0.23$ | $7.95 \pm 0.05$ | $\mathbf{7.68} \pm 0.07$ | $7.91 \pm 0.08$ | $7.90 \pm 0.13$ |
| ResNet-56 | 6.54 | $7.97 \pm 0.22$ | $7.70 \pm 0.06$ | $\mathbf{7.52} \pm 0.18$ | $7.60 \pm 0.09$ | $7.61 \pm 0.12$ |

## 4 Experiments

We consider two important tasks for regularized training in deep learning communities: (i) training sparse neural networks and (ii) network quantization. Throughout our experiments, we consider ADAM as a representative of PROXGEN where $m_t = \rho_t m_{t-1} + (1 - \rho_t)g_t$ with constant decaying parameter $\rho_t = 0.9$ and $C_t = \sqrt{\beta C_{t-1} + (1 - \beta)g_t^2}$ with $\beta = 0.999$ in Algorithm 1. The details on other hyperparameter/experiment settings are provided in the Appendix.

**Training Sparse Neural Networks.** Motivated by the lottery ticket hypothesis [50], we consider training VGG-16 [51] and ResNet-34 [52] on CIFAR-10 dataset using sparsity encouraging regularizers. Toward this, we consider the following objective function with possibly non-convex $\ell_q$ regularization: $F(\theta) := \mathbb{E}_{\xi \sim \mathbb{P}}[f(\theta; \xi)] + \lambda \sum_{j=1}^{p} |\theta_j|^q$ where $0 \leq q \leq 1$. We train the network parameters with the closed-form proximal mappings introduced in Section 2. The results on VGG-16 are provided in Appendix.

We compare PROXGEN with subgradient methods and also include PROX-SGD [16] as a baseline especially for $\ell_1$ regularization since PROX-SGD considers only convex regularizers. In PROX-SGD, the hand-crafted fine-tuned scheduling on $\alpha_t$ and $\rho_t$ is essential for fast convergence and good performance, but in our experiments we use standard settings $\rho_t = 0.9$. We first validate our theory in practice using constant stepsize in order to purely see the effect of proximal approaches (the results on this setting are provided in the Appendix). Then the step-decay learning rate scheduling is employed to consider standard training schemes for the state-of-the-art performance, which also satisfies the non-increasing stepsize condition in our Theorem 1. For $\ell_0$ regularization, the problem in (1) cannot be optimized in a subgradient manner, so we compare PROXGEN with another popular baseline, $\ell_{0_{hc}}$ [6] which approximates the $\ell_0$-norm via hard-concrete distributions.

Figures 2 illustrates the results for ResNet-34. In terms of convergence, PROXGEN shows faster convergence than PROX-SGD for $\ell_1$ case in Figure 2-(a), but there is no difference between PROXGEN and subgradient methods as in Figure 2-(a). However, there are notable differences in convergence for non-convex regularizers $\ell_{1/2}$ and $\ell_{2/3}$, which get bigger as $q$ decreases. We believe this might be because the $\ell_q$-norm derivative, $q/|\theta|^{1-q}$, is very large for non-zero tiny $\theta$ for $q \in (0, 1)$. Meanwhile, $\partial|\theta|/\partial\theta$ is merely the sign value regardless of size of $\theta$, so the large gradient of $|\theta|^q$ may hinder convergence. The learning curves in Figure 2-(b,c) empirically corroborate this phenomenon.

In terms of performance, we can see that PROXGEN consistently achieves better performance than baselines for ResNet-34 with similar or even better sparsity level. Importantly, PROXGEN with $\ell_0$ outperforms $\ell_{0_{hc}}$ baseline by a great margin. This might be due to the design of $\ell_{0_{hc}}$, which approximates $\|\theta\|_0 = \sum_{j=1}^{p} \mathbb{I}\{\theta_j \neq 0\}$ with binary mask $z_j$ parameterized by learnable probability $\pi_j$ for each coordinate. Thus, the number of parameters to be optimized is doubled, which might make optimization harder. In constrast, PROXGEN does not introduce additional parameters.

More results for other famous non-convex regularizers MCP [53] and SCAD [54] are in Appendix.

**Training Group-Sparse Neural Networks.** In the Appendix, we consider training Statistical Recurrent Units where $\ell_{1,2}$ group-norm penalty is imposed on the input layer weights to detect non-linear Granger Causality [55]. As the proximal mappings for PROXGEN with group sparsity are not available in closed-form, we develop an efficient procedure for computation, whose derivations are also provided in the Appendix.

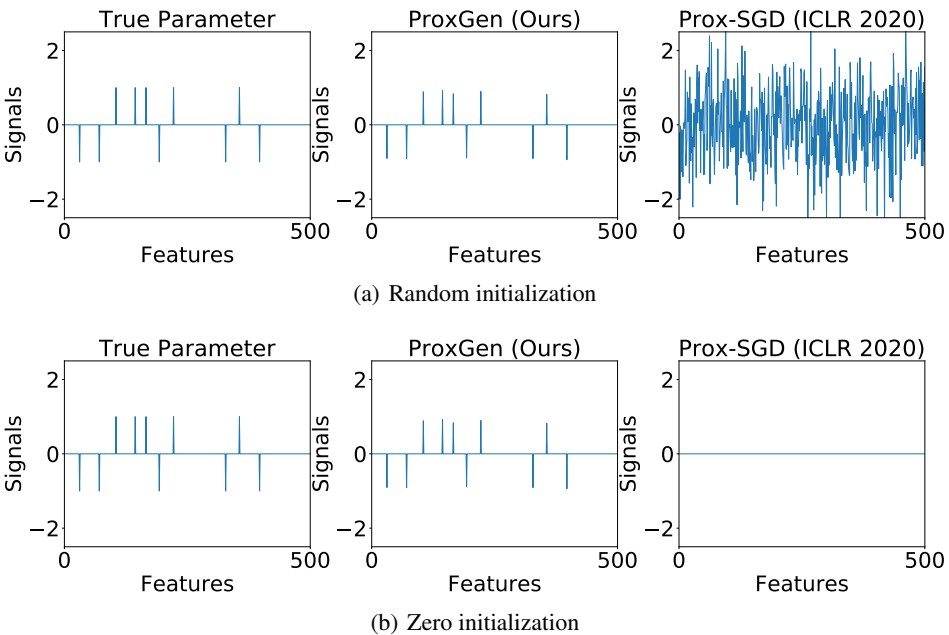

(a) Random initialization

(b) Zero initialization

Figure 3: Lasso simulations with different initialization schemes.

**Training Binary Neural Networks.** We consider the network quantization constraining the parameters to some set of discrete values which is a key approach for model compression. We evaluate our revised PROXQUANT in Table 2 with extended regularization $\mathcal{R}^q_{\text{bin}}$ in Section 2. We consider the following objective function with quantization-specific regularizers: $F(\theta) := \mathbb{E}_{\xi \sim \mathbb{P}}[f(\theta; \xi)] + \lambda \sum_{j=1}^p |\theta_j - \text{sign}(\theta_j)|^q$ where $0 \le q \le 1$. For comparisons, we quantize ResNet weight parameters (except bias and activations) on CIFAR-10 and ImageNet dataset.

Table 3 presents the results. For all $q$ values, revised PROXQUANT consistently outperforms the baselines except for ResNet-20, which implies PROXGEN may work better for larger networks. As such, our generalized regularizers $\mathcal{R}^q_{\text{bin}}$ contribute to one of the state-of-the-art optimization-based methods in network quantization. Notably, revised PROXQUANT $\ell_1$ greatly outperforms PROXQUANT baseline while these two approaches differ only in update rules (see Table 2). Hence, we can conclude that revised PROXQUANT based on PROXGEN provides an *exact* proximal update and also yields more generalizable solutions. In our experience, revised PROXQUANT $\ell_0$ shows little degradation in performance, so we do not include this result. However, revised PROXQUANT $\ell_0$ shows superiority to baselines for language modeling, whose preliminary results are in Appendix.

Table 4: Comparison for binary neural networks for ImageNet. [†] means the first and last layer not quantized.

| | | ResNet-18 | |
| --- | --- | --- | --- |
| | | Top-1 Error (%) | Top-5 Error (%) |
| Full precision | | 30.46 | 10.81 |
| BWN [56] | | 39.20 | 17.00 |
| LR-Net[†] [57] | | 40.10 | 17.70 |
| ELQ [58] | | 35.28 | 13.96 |
| PROXQUANT [1] | | 36.24 | 14.23 |
| Revised PROXQUANT $\ell_1$ (Ours) | | **34.85** | **12.38** |

Table 4 illustrates Top-1/Top-5 error (%) for training ResNet on ImageNet with binary quantization. The most important thing is that our revised PROXQUANT shows great improvements in performance over the original PROXQUANT. Furthermore, PROXGEN shows superior performance to various baselines for weight quantization.

## 5 A Closer Look into Prox-SGD [16] vs. PROXGEN

Prox-SGD [16] is the approach closest to our PROXGEN method. However, PROX-SGD is *not an exact* proximal approach and is significantly different from PROXGEN. PROXGEN's update rule

involves directly solving the quadratic subproblem (2). In contrast, PROX-SGD's update rule consists of two stages: (i) solving the quadratic subproblem *without* learning rate (5), then (ii) updating the parameters with the computed direction (i.e. $\widehat{\theta}_t - \theta_t$) by the learning rate $\alpha_t$ (6).

$$\widehat{\theta}_t = \underbrace{\mathrm{prox}_{\lambda\mathcal{R}(\cdot)}^{C_t+\delta I}\left(\theta_t - (C_t+\delta I)^{-1}m_t\right)}_{\text{no learning rate}},\tag{5}$$

$$\theta_{t+1} = \theta_t + \alpha_t(\widehat{\theta}_t - \theta_t)\tag{6}$$

To clearly see the differences between both approaches, we conduct two studies.

**Study 1: Lasso Support Recovery.** For this task, the two-stage update scheme of PROX-SGD might have some potential issues. For example, for $\ell_1$-regularized problems, the updated parameter $\theta_{t+1}$ (6) *might not achieve exact zero* (while $\widehat{\theta}_t$ can) whereas $\theta_{t+1}$ for PROXGEN (2) can attain exact zero value according to the update rule (3) in Section 2. Another potential caveat is that PROX-SGD might *overestimate* the sparsity level. In view of the above, we run Lasso simulations with different two initialization schemes: (i) random initialization and (ii) zero initialization. For random initialization, it can be seen in Figure 3-(a) that PROX-SGD could not achieve exact zero value, which corroborates our first observation. More interestingly, for zero initialization, we can see in Figure 3-(b) that the estimates using PROX-SGD are exactly zeros for all coordinates, which supports our second observation. This might be because $\widehat{\theta}_t$ (5) is always zero since the quadratic subproblem does not consider the learning rate, which might overestimate the sparsity level. Hence, the subsequent iterate $\theta_{t+1}$ would be always zero since we initialize the parameters with zeros, but PROXGEN recovers the correct support in both cases.

**Study 2: DenseNet-201 on CIFAR-100 Dataset.** To validate the superiority of PROXGEN upon PROX-SGD, we revisit the largest experiments in [16]. We train DenseNet-201 architecture on CIFAR-100 dataset with $\ell_1$ regularization since PROX-SGD only consider convex regularizers. For both methods, we use the same hyperparameter settings for fair comparison. Figure 4 illustrates the training learning curves, and it can be seen that our PROXGEN achieves faster convergence as well as lower objective values. For our experience, the learning curves show the similar dynamics for different $\lambda$ values.

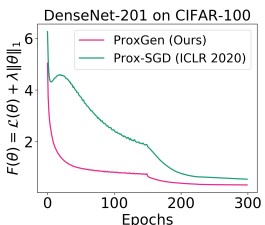

Figure 4: Learning curve.

**Comparison of Theoretical Contributions.** [16] guarantees the convergence of PROX-SGD, but *not how fast* it converges. Moreover, this is proved *without* considering preconditioners. In contrast, our analysis for the PROXGEN framework appropriately incorporates the first-order momentum and arbitrary positive preconditioner with detailed *non-asymptotic* convergence.

# 6 Conclusion

In this work, we proposed PROXGEN, the first general family of stochastic proximal gradient methods. Within our framework, we presented novel examples of proximal versions of standard SGD approaches, including a proximal version of ADAM. We analyzed the convergence of the whole PROXGEN family and showed that PROXGEN can encompass the results of several previous studies. We also demonstrated that PROXGEN empirically outperforms subgradient-based methods for popular deep learning problems. As future work, we plan to further study efficient procedures to compute the proximal mappings for structured regularizers such as $\ell_1/\ell_q$-norms with preconditioners.

# Acknowledgement

This work was supported by the National Research Foundation of Korea (NRF) grants (2018R1A5A1059921, 2019R1C1C1009192) and Institute of Information & Communications Technology Planning & Evaluation (IITP) grants (No.2019-0-01371, Development of brain-inspired AI with human-like intelligence, No.2019-0-00075, Artificial Intelligence Graduate School Program (KAIST)) funded by the Korea government (MSIT).

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
