## Supplementary Materials

## A  Sparse Neural Networks with $\ell_q$ Regularization with Constant Stepsize

We include the experimental results on sparse neural networks using ResNet-34 and constant stepsize in Figure 5. As seen in Figure 5, the proximal methods in this regime also shows superior performance than the subgradient baselines.

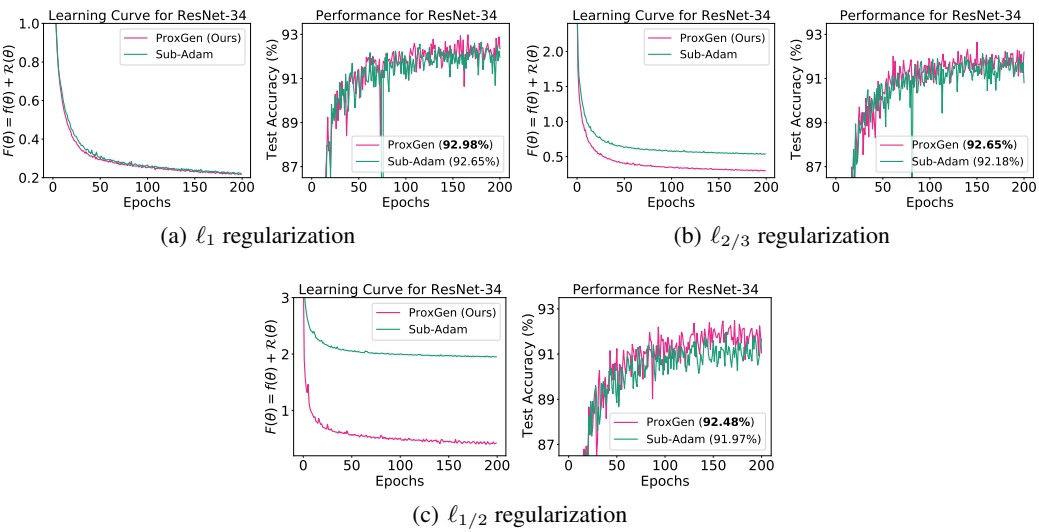

(a) $\ell_1$ regularization                                       (b) $\ell_{2/3}$ regularization

(c) $\ell_{1/2}$ regularization

Figure 5: Comparison for sparse ResNet-34 on CIFAR-10 dataset using constant stepsize.

## B  Sparse Neural Networks with $\ell_q$ Regularization for VGG-16

We include the experimental results on sparse neural networks using VGG-16 architecture in Figure 6.

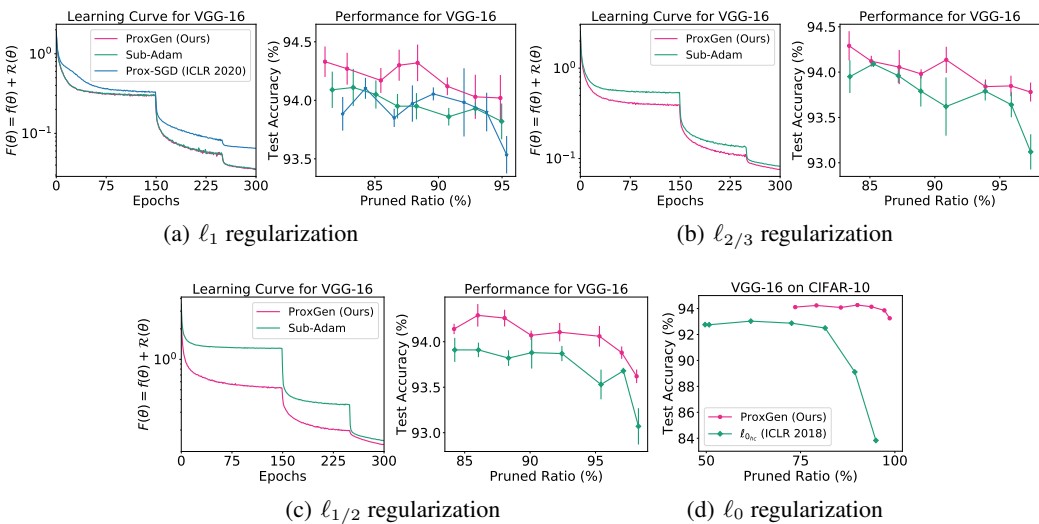

(a) $\ell_1$ regularization                                       (b) $\ell_{2/3}$ regularization

(c) $\ell_{1/2}$ regularization                                   (d) $\ell_0$ regularization

Figure 6: Comparison for sparse VGG-16 on CIFAR-10 dataset.

Table 5: Configuration and Hyperparameters for the SRU experiments.

| Parameters | Dataset | | | | |
|---|---|---|---|---|---|
| | Lorenz F=10 | Lorenz F=40 | VAR | Dream-3 | NetSim |
| Learning rate | 0.005 | 0.01 | 0.04 | 0.005 | 0.001 |
| Batch size | 125 | 125 | 125 | 21 | 5 |
| # Training epochs | 2000 | 2000 | 2000 | 1000 | 2000 |
| # units per layer | 10 | | | | |
| $\mathcal{A}$ | $\{0.0, 0.01, 0.1, 0.99\}$ | | | | |
| Group-sparse reg. param. for the input layer | $[0.01, 10]$ | | | | |

## C  Additional Experiments: Sparse Neural Networks with MCP and SCAD Non-convex Regularizers

We provide the additional experiments for sparse neural networks with MCP [53] and SCAD [54] non-convex regularizers. Figure 7 and 8 illustrate the results for VGG-16 and ResNet-34 respectively. As shown in Section 4 and these figures, PROXGEN is very effective for solving the non-convex regularized problems.

## D  Experiments on Group-Sparse Neural Networks

In this section, we consider estimating group-sparse Neural networks for the task of detecting non-linear Granger Causality in multivariate time series data. Granger causality [59] is a widely used approach for time series structure discovery. Several approaches have been proposed recently that employ structured multilayer perceptrons (MLPs) or recurrent neural networks (RNNs) [60, 55]. Following [55] (Section 3), given $n$ time series, for each time series $x_j$, we consider training a Statistical Recurrent Unit network to predict the future value $x_j$ based on the past value of all the $n$ time series. The $\ell_{1,2}$ group-norm penalty is imposed on the input layer parameters where group $G_i$ is formed by the input layer parameters corresponding to input time series $x_i$. Then, time series $x_i$ is detected as Granger-causing time series $x_j$ if the input layer parameters in group $G_i$ for the SRU network predicting $x_j$ are non-zero.

**Comparison methods.**   For optimization of the group-sparsity regularized SRUs, [55] employs ADAM where each ADAM step is followed by a proximal step via group soft-thresholding of the input layer parameters. This proximal step uses ADAM's initial learning rate and disregards its coordinate-wise adaptivity, hence the optimization method of [55] is not a "pure" proximal stochastic gradient descent approach. In this section, we compare the algorithm of [55](using the implementation from https://github.com/sakhanna/SRU_for_GCI) with PROXGEN. The proximal updates for PROXGEN with the $\ell_{1,2}$ group-norm penalty are not available in closed-form and we use the procedure described in Section F to compute them.

**Evaluation metrics.**   We follow the same experimental setup as in [55] and evaluate the methods in terms of their accuracy in detecting causal links among time series. Specifically we report the AUROC (Area Under the Receiver Operating Characteristic curve), where the ROC curve illustrates the trade off between the true-positive rate (TPR) and the false-positive rate (FPR) achieved by the methods towards the detection of pairwise Granger causal relationships.

**Datasets.**   We use the same datasets as in [55] : *Lorenz* (F=10/40) where $T = \{250, 500\}$ measurements for 10 time series variables are generated according to the Lorenz-96 model and $F$ denotes the magnitude of the external forcing in the model; *VAR* simulations for a 3rd order VAR model with 10 components and $T = \{500, 1000\}$ measurements ; *NetSim* where we use $T = 200$ time-ordered signals that are simulated for 15 brain regions in 5 human subjects labelled $2 - 6$; and *Dream-3 (E.coli-1)* where gene expression levels for 100 genes are measured for *E.coli* over 966 time points. The datasets are all available from https://github.com/sakhanna/SRU_for_GCI.

**Hyperparameters and configuration.**   We use the same hyperparameters and SRU configuration as [55] (Table 10). The only difference is that for PROGEN we set the range of the penalty parameter for the group penalty to $[0.001, 10]$. The setups are summarized in Table 5.

**Results.**   The AUROC for the various datasets are presented in Table 6. As can be seen from the table, PROXGEN achieves higher AUROC in most cases. We believe that the improved performance of PROXGEN is

Table 6: Comparison for group-sparsity regularized SRUs. AUROC (the higher the better) for detecting pairwise causal links from multivariate time series data.

| | [55] | PROXGEN |
|---|---|---|
| Lorenz ($F = 10, T = 250$) | $0.83 \pm 0.03$ | $\mathbf{0.94 \pm 0.02}$ |
| Lorenz ($F = 10, T = 500$) | $0.90 \pm 0.02$ | $\mathbf{0.98 \pm 0.05}$ |
| Lorenz ($F = 40, T = 250$) | $\mathbf{1.00 \pm 0.00}$ | $\mathbf{1.00 \pm 0.00}$ |
| Lorenz ($F = 40, T = 500$) | $\mathbf{1.00 \pm 0.00}$ | $\mathbf{1.00 \pm 0.00}$ |
| VAR (T=500) | $0.82 \pm 0.06$ | $\mathbf{0.88 \pm 0.04}$ |
| VAR (T=1000) | $0.91 \pm 0.04$ | $\mathbf{0.93 \pm 0.05}$ |
| NetSim | $\mathbf{0.79 \pm 0.03}$ | $0.78 \pm 0.02$ |
| Dream-3 (E.coli-1) | $0.657$ | $\mathbf{0.660}$ |

directly attributable to the incorporation of the preconditionners in the proximal mappings. As future work, we plan to experiment with other architectures beyond SRUs, such as the Economy SRUs proposed in [55].

## E  Details on Experimental Settings

**Sparse Neural Networks.**  To reflect the most practical training settings, we first tune the weight-decay parameter $\zeta$ without $\ell_q$ regularizers.  For weight-decay coefficients, we consider the candidates $\zeta \in \{0.001, 0.002, 0.005, 0.01, 0.02, 0.05, 0.1, 0.2, 0.5\}$ for $\zeta$ and the best $\zeta$ value is 0.2 for both networks VGG-16 and ResNet-34 in our experience. After tuning weight-decay coefficient $\zeta$, we consider both decoupled weight decay [61] and $\ell_q$ regularization whose detail update rule is described in Algorithm 2.  For all comparison methods except $\ell_{0_{hc}}$, the recommended stepsize $\alpha_t = 0.001$ is employed, but we tune this stepsize for $\ell_{0_{hc}}$ baseline.  We consider a broad range of regularization parameters for all methods: $\lambda \in \{0.001, 0.002, 0.005, 0.01, 0.02, \cdots, 1.0, 2.0, 5.0\}$. With these hyperparameter settings, we consider the total 300 epochs and divide the learning rate at 150-th and 250-th epoch by 10.

**Binary Neural Networks.**  In this experiment, we follow the same experimental settings in baseline PROXQUANT [1].  We first pre-train ResNet-{20, 32, 44, 56} with full-precision and initialize the network parameters with these pre-trained weights.  Then, we consider the total 300 epochs and hard-quantize the networks at 200-th epoch (i.e. quantizing the weight parameters to $+1$ or $-1$). We employ the homotopy method introduced in [1]: annealing the regularization paramter $\lambda$ as $\lambda_{\text{epoch}} = \lambda \times$ epoch. For initial value of $\lambda$, we use $\lambda = 10^{-8}$ or $\lambda = 5 \cdot 10^{-8}$ for all ResNet architecture. We use the constant stepsize $\alpha_t = 0.01$ as recommended in [1].

**Lasso Support Recovery.**  We generate simple Lasso simulations with problem dimension $p = 500$ and $n = 100$ data samples. The number of non-zero entries in true parameter vector $\theta^* \in \mathbb{R}^p$ is set to 10. The design matrix $X \in \mathbb{R}^{n \times p}$ is generated from standard Gaussian distribution $\mathcal{N}(0, 1)$ and we randomly assign $+1$ or $-1$ for the non-zero value in true parameter at random 10 coordinates. The response variable $y \in \mathbb{R}^n$ is generated with small noise by $y = X\theta^* + \epsilon$ where $\epsilon \sim \mathcal{N}(0, 0.05^2)$. For both PROXGEN and PROX-SGD, we employ ADAM for preconditioner matrix $C_t$ construction.

Here, we introduce preliminary results of revised PROXQUANT $\ell_0$ on language modeling. For this experiment, we train one hidden layer LSTM with embedding dimension 300 and 300 hidden units according to [1]. First, we pre-train the full-precision LSTM and initialize the network with pre-trained weights. We consider the total 80 epochs and divide the learning rate by 1.2 if the validation loss does not decrease. Table 7 shows the preliminary results and revised PROXQUANT $\ell_0$ is superior to the PROXQUANT baseline in this task.

Table 7: Preliminary results on revised PROXQUANT $\ell_0$ for LSTM models.

| Algorithm | Test Perplexity |
|---|---|
| Full-precision (32-bit) | 88.5 |
| BinaryConnect [8] | 372.2 |
| PROXQUANT [1] | 288.5 |
| revised PROXQUANT $\ell_0$ (Ours) | **223.4** |

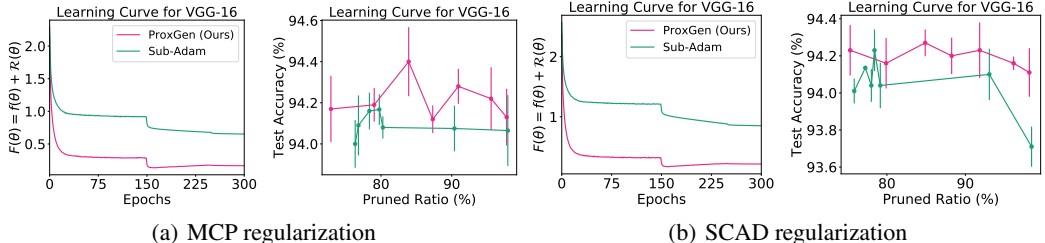

(a) MCP regularization

(b) SCAD regularization

Figure 7: Comparison for sparse VGG-16 on CIFAR-10 dataset with other non-convex regularizers.

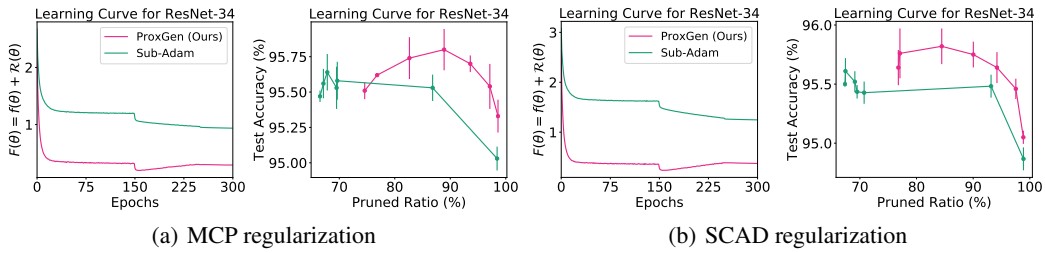

(a) MCP regularization

(b) SCAD regularization

Figure 8: Comparison for sparse ResNet-34 on CIFAR-10 dataset with other non-convex regularizers.

## F    Derivations for Proximal Mappings

We first derive the concrete update rule for $\ell_q$ regularization with *diagonal* preconditioners as introduced in Section 2. Next, we provide closed-form proximal mappings for MCP and SCAD regularization. Finally, we consider group $\ell_{1,2}$ regularization.

$\ell_{1/2}$ **regularization.**    First, we review the closed-form proximal mappings for $\ell_{1/2}$ regularization of vanilla SGD. First, we consider the following one-dimensional program:

$$\widehat{x} = \operatorname*{argmin}_{x}\{(x - z)^2 + \lambda|x|^{1/2}\} \tag{7}$$

For the program (7), it is known that the closed-form solution exists [37] as

$$\widehat{x} = \begin{cases} \frac{2}{3}|z|\left(1 + \cos\left(\frac{2}{3}\pi - \frac{2}{3}\varphi_\lambda(z)\right)\right) & \text{if } z > p(\lambda) \\ 0 & \text{if } |z| \le p(\lambda) \\ -\frac{2}{3}|z|\left(1 + \cos\left(\frac{2}{3}\pi - \frac{2}{3}\varphi_\lambda(z)\right)\right) & \text{if } z < -p(\lambda) \end{cases} \tag{8}$$

where $\varphi_\lambda(z) = \arccos\left(\frac{\lambda}{8}\left(\frac{|z|}{3}\right)^{-3/2}\right)$ and $p(\lambda) = \frac{\sqrt[3]{54}}{4}(\lambda)^{2/3}$. Based on this closed-form solution, we derive PROXGEN for $\ell_{1/2}$ regularization with diagonal preconditioners. By (2), we have

$$\widehat{\theta}_t = \theta_t - \alpha_t(C_t + \delta I)^{-1}m_t \tag{9}$$

$$\theta_{t+1} \in \operatorname{prox}_{\alpha_t\lambda\mathcal{R}(\cdot)}^{C_t+\delta I}(\widehat{\theta}_t) \tag{10}$$

$$= \operatorname*{argmin}_{\theta}\left\{\frac{1}{2}\|\theta - \widehat{\theta}_t\|^2_{C_t+\delta I} + \lambda\sum_{j=1}^{p}|\theta_j|^{1/2}\right\} \tag{11}$$

Since the program (11) is coordinate-wise decomposable (since the preconditioner matrix $C_t$ is diagonal), we can split (11) into

$$\theta_{t+1,i} = \operatorname*{argmin}_{\theta_i}\left\{\frac{1}{2}(C_{t,i} + \delta)(\theta_i - \widehat{\theta}_{t,i})^2 + \alpha_t\lambda|\theta_i|^{1/2}\right\}$$

$$= \operatorname*{argmin}_{\theta_i}\left\{(\theta_i - \widehat{\theta}_{t,i})^2 + \frac{2\alpha_t\lambda}{C_{t,i} + \delta}|\theta_i|^{1/2}\right\}$$

for the $i$-th coordinate. From (7), we can derive

$$\theta_{t+1,i} = \begin{cases} \frac{2}{3}|\widehat{\theta}_{t,i}|\left(1 + \cos\left(\frac{2}{3}\pi - \frac{2}{3}\varphi_\lambda(\widehat{\theta}_{t,i})\right)\right) & \text{if } \widehat{\theta}_{t,i} > p(\lambda) \\ 0 & \text{if } |\widehat{\theta}_{t,i}| \leq p(\lambda) \\ -\frac{2}{3}|\widehat{\theta}_{t,i}|\left(1 + \cos\left(\frac{2}{3}\pi - \frac{2}{3}\varphi_\lambda(\widehat{\theta}_{t,i})\right)\right) & \text{if } \widehat{\theta}_{t,i} < -p(\lambda) \end{cases}$$

where

$$\varphi_\lambda(\widehat{\theta}_{t,i}) = \arccos\left(\frac{\alpha_t\lambda}{4(C_{t,i}+\delta)}\left(\frac{|\widehat{\theta}_{t,i}|}{3}\right)^{-3/2}\right), \quad p(\lambda) = \frac{\sqrt[3]{54}}{4}\left(\frac{2\alpha_t\lambda}{C_{t,i}+\delta}\right)^{2/3}.$$

$\ell_{2/3}$ **regularization.** Now, we provide the closed-form solutions for proximal $\ell_{2/3}$ mappings with diagonal preconditioners. Similar to $\ell_{1/2}$ regularization, we start from the closed-form solutions of the following program:

$$\widehat{x} = \underset{x}{\operatorname{argmin}}\{(x-z)^2 + \lambda|x|^{2/3}\} \tag{12}$$

The closed-form solution for the program (12) is known to be

$$\widehat{x} = \begin{cases} \left(\frac{|A|+\sqrt{\frac{2|z|}{|A|}-|A|^2}}{2}\right)^3 & \text{if } z > \frac{2}{3}\sqrt[4]{3\lambda^3} \\ 0 & \text{if } |z| \leq \frac{2}{3}\sqrt[4]{3\lambda^3} \\ -\left(\frac{|A|+\sqrt{\frac{2|z|}{|A|}-|A|^2}}{2}\right)^3 & \text{if } z < -\frac{2}{3}\sqrt[4]{3\lambda^3} \end{cases} \tag{13}$$

where

$$|A| = \frac{2}{\sqrt{3}}\lambda^{1/4}\left(\cosh\left(\frac{\phi}{3}\right)\right)^{1/2}, \quad \phi = \operatorname{arccosh}\left(\frac{27z^2}{16}\lambda^{-3/2}\right) \tag{14}$$

Based on this formulation, we derive the closed-form proximal mappings with diagonal preconditioner $C_t$. By (2), we have

$$\widehat{\theta}_t = \theta_t - \alpha_t(C_t + \delta I)^{-1}m_t \tag{15}$$

$$\theta_{t+1} \in \operatorname{prox}_{\alpha_t\lambda\mathcal{R}(\cdot)}^{C_t+\delta I}(\widehat{\theta}_t) \tag{16}$$

$$= \underset{\theta}{\operatorname{argmin}}\left\{\frac{1}{2}\|\theta - \widehat{\theta}_t\|_{C_t+\delta I}^2 + \lambda\sum_{j=1}^{p}|\theta_j|^{2/3}\right\} \tag{17}$$

As in $\ell_{1/2}$ case, the program (17) is coordinate-wise separable, so it suffices to solve the sub-problems for each coordinate as

$$\theta_{t+1,i} = \underset{\theta_i}{\operatorname{argmin}}\left\{\frac{1}{2}(C_{t,i}+\delta)(\theta_i - \widehat{\theta}_i)^2 + \alpha_t\lambda|\theta_i|^{2/3}\right\}$$

$$= \underset{\theta_i}{\operatorname{argmin}}\left\{(\theta_i - \widehat{\theta}_{t,i})^2 + \frac{2\alpha_t\lambda}{C_{t,i}+\delta}|\theta_i|^{2/3}\right\}$$

From (12), we can derive

$$\theta_{t+1,i} = \begin{cases} \left(\frac{|A|+\sqrt{\frac{2|\widehat{\theta}_{t,i}|}{|A|}-|A|^2}}{2}\right)^3 & \text{if } \widehat{\theta}_{t,i} > \frac{2}{3}\sqrt[4]{3\lambda^3} \\ 0 & \text{if } |\widehat{\theta}_{t,i}| \leq \frac{2}{3}\sqrt[4]{3\lambda^3} \\ -\left(\frac{|A|+\sqrt{\frac{2|\widehat{\theta}_{t,i}|}{|A|}-|A|^2}}{2}\right)^3 & \text{if } \widehat{\theta}_{t,i} < -\frac{2}{3}\sqrt[4]{3\lambda^3} \end{cases}$$

where

$$|A| = \frac{2}{\sqrt{3}}\left(\frac{2\alpha_t\lambda}{C_{t,i}+\delta}\right)^{1/4}\left(\cosh\left(\frac{\phi}{3}\right)\right)^{1/2}, \quad \phi = \operatorname{arccosh}\left(\frac{27\widehat{\theta}_{t,i}^2}{16}\left(\frac{2\alpha_t\lambda}{C_{t,i}+\delta}\right)^{-3/2}\right)$$

In addition to $\ell_q$ regularization, we provide the closed-form proximal mappings for MCP and SCAD regularizers with non-trivial preconditioners.

**MCP regularization.**  Before introducing the closed-form of proximal m mappings for MCP regularized problems with diagonal preconditioners, we first review the MCP regularizer. The MCP regularizer is defined as

$$\rho_\lambda(x;b) = \begin{cases} \lambda|x| - \frac{x^2}{2b} & \text{if } |x| \leq b\lambda \\ \frac{b\lambda^2}{2} & \text{if } |x| > b\lambda \end{cases} \tag{18}$$

where $b > 0$ is called the MCP parameter and $\lambda$ is a regularization parameter. Our goal is to derive the proximal mapping of this regularizer with diagonal preconditioner.

Now, we start from the closed-form solutions of the following program:

$$\widehat{x} = \underset{x}{\operatorname{argmin}}\{\frac{1}{2}(x-z)^2 + \rho_\lambda(x;b)\} \tag{19}$$

For this program, the closed-form solution is known as

$$\widehat{x} = \operatorname{sign}(z) \min\left\{ \frac{b\max\{|z|-\lambda, 0\}}{b-1}, |z| \right\} \tag{20}$$

Based on this closed-form solution, we derive the closed-form proximal mappings with diagonal preconditioner $C_t$. By (2), we have

$$\widehat{\theta}_t = \theta_t - \alpha_t(C_t + \delta I)^{-1}m_t \tag{21}$$

$$\theta_{t+1} \in \operatorname{prox}_{\alpha_t\rho_\lambda(\cdot;b)}^{C_t+\delta I}(\widehat{\theta}_t) \tag{22}$$

$$= \underset{\theta}{\operatorname{argmin}}\left\{ \frac{1}{2}\|\theta - \widehat{\theta}_t\|_{C_t+\delta I}^2 + \alpha_t\rho_\lambda(\theta;b) \right\} \tag{23}$$

Since this program is also coordinate-wise separable, we could have for each coordinate

$$\theta_{t+1,i} = \operatorname{sign}(\widehat{\theta}_{t,i}) \min\left\{ \frac{b\max\{|\widehat{\theta}_{t,i}| - \frac{\alpha_t\lambda}{C_{t,i}+\delta}, 0\}}{b-1}, |\widehat{\theta}_{t,i}| \right\} \tag{24}$$

**SCAD regularization.**  We first introduce SCAD regularizer defined as :

$$\rho_\lambda(x;a) = \begin{cases} \lambda|x| & \text{if } |x| \leq \lambda \\ \frac{-\lambda^2 - 2a\lambda|x| + x^2}{2(a-1)} & \text{if } \lambda < |x| \leq a\lambda \\ \frac{(a+1)\lambda^2}{2} & \text{if } |x| > a\lambda \end{cases} \tag{25}$$

where $a > 2$ is called the SCAD parameter and $\lambda$ is a regularization parameter. As in MCP regularizer, we start from the following program

$$\widehat{x} = \underset{x}{\operatorname{argmin}}\left\{ \frac{1}{2}\|x-z\|^2 + \rho_\lambda(x;a) \right\}$$

The closed-form solution for this program is known as

$$\widehat{x} = \begin{cases} \operatorname{sign}(z)\max\{|z|-\lambda, 0\} & \text{if } |z| \leq 2\lambda \\ \frac{(a-1)z - \operatorname{sign}(z)a\lambda}{a-2} & \text{if } 2\lambda < |z| \leq a\lambda \\ z & \text{if } |z| > a\lambda \end{cases} \tag{26}$$

Based on this formulation, we could derive the closed-form solution for PROXGEN with diagonal preconditioner. By (2), we have

$$\widehat{\theta}_t = \theta_t - \alpha_t(C_t + \delta I)^{-1}m_t \tag{27}$$

$$\theta_{t+1} \in \operatorname{prox}_{\alpha_t\rho_\lambda(\cdot;a)}^{C_t+\delta I}(\widehat{\theta}_t) \tag{28}$$

$$= \underset{\theta}{\operatorname{argmin}}\left\{ \frac{1}{2}\|\theta - \widehat{\theta}_t\|_{C_t+\delta I}^2 + \alpha_t\rho_\lambda(\theta;a) \right\} \tag{29}$$

Since the program is coordinate-wise decomposable, we have for each coordinate

$$\theta_{t+1,i} = \begin{cases} \operatorname{sign}(\widehat{\theta}_{t,i})\max\{|\widehat{\theta}_{t,i}| - \widehat{\lambda}_i|, 0\} & \text{if } |\widehat{\theta}_{t,i}| \leq 2\widehat{\lambda}_i \\ \frac{(a-1)\widehat{\theta}_{t,i} - \operatorname{sign}()\widehat{\theta}_{t,i}a\widehat{\lambda}_i}{a-2} & \text{if } 2\widehat{\lambda}_i < |\widehat{\theta}_{t,i}| \leq a\widehat{\lambda}_i \\ \widehat{\theta}_{t,i} & \text{if } |\widehat{\theta}_{t,i}| > a\widehat{\lambda}_i \end{cases} \tag{30}$$

where $\widehat{\lambda}_i = \frac{\alpha_t\lambda}{C_{t,i}+\delta}$.

Although the derivations look little complicated for both cases, we emphasize that both two closed-form solutions can be efficiently implemented in a GPU-friendly manner.

**Group $\ell_{1,2}$ regularization.** When there is no preconditioning, the proximal mapping for the $\ell_{1,2}$ penalty can be computed in closed-form via group soft-thresholding. In the presence of preconditioners, the proximal mapping for the $\ell_{1,2}$ group penalty is no longer available in closed-form, but can be computed easily as follows.

Let $\theta$ denote the network parameters that are being regularized via group penalty, let $\{G_1, \ldots, G_K\}$ denote their partition into $K$ groups, and let $\theta_{(k)}$ denote the subset of the parameters corresponding to group $G_k$. The proximal mapping for the $\ell_{1,2}$ group-norm penalty is

$$\text{prox}^{\ell_{1,2}}_{C_t+\delta I}(\theta_t) = \underset{\theta}{\text{argmin}} \left\{ \frac{1}{2}(\theta - \theta_t)^T (C_t + \delta I)(\theta - \theta_t) + \alpha_t \lambda \sum_{k \in K} \|\theta_{(k)}\|_2 \right\} \tag{31}$$

where $C_t$ is a diagonal matrix.

The problem is separable with respect to the groups. For each group $G_k$ we have to solve

$$\text{prox}^{\ell_{1,2}}_{D_{t,(k)}}(\theta_{t,(k)}) = \underset{\theta_{(k)}}{\text{argmin}} \left\{ \frac{1}{2}(\theta_{(k)} - \theta_{t,(k)})^T D_{t,(k)}(\theta_{(k)} - \theta_{t,(k)}) + \alpha_t \lambda \|\theta_{(k)}\|_2 \right\}, \tag{32}$$

where $D_{t,(k)} = C_{t,(k)} + \delta I_{(k)}$.

The solution is provided in the following Lemma.

**Lemma 1.** *The solution to* (32) *is given by*

$$\text{prox}^{\ell_{1,2}}_{D_{t,G_k}}(\theta_{t,G_k}) = \begin{cases} 0, & \|D_{t,(k)}\theta_{t,(k)}\|_2 \leq \alpha_t \lambda, \\ \widetilde{D}_{t,(k)}\theta_{t,(k)}, & \|D_{t,(k)}\theta_{t,(k)}\|_2 > \alpha_t \lambda \end{cases}, \tag{33}$$

*where $\widetilde{D}_{t,(k)}$ is a diagonal matrix with diagonal entries given by*

$$[\widetilde{D}_{t,(k)}]_{ii} = \frac{[D_{t,(k)}]_{ii}}{[D_{t,(k)}]_{ii} + \alpha_t \lambda/\xi},$$

*and $\xi$ is defined as the unique solution to*

$$1 = \sum_{i \in G_k} \left( \frac{[D_{t,(k)}]_{ii}[\theta_{t,(k)}]_i}{\xi[D_{t,(k)}]_{ii} + \alpha_t \lambda} \right)^2. \tag{34}$$

*Proof.* The proximal problem is a strongly convex quadratic that has a solution. If the solution is non-zero, the objective function is differentiable, and the solution satisfies

$$0 = D_{t,(k)}(\theta_{(k)} - \theta_{t,(k)}) + \alpha_t \lambda \frac{\theta_{(k)}}{\|\theta_{(k)}\|_2}$$

from which the form of $\widetilde{D}_{t,(k)}$ immediately follows. Plugging the above optimality condition into (32) results in a scalar problem in $\xi = \|\theta_{(k)}\|_2$:

$$\min_{\xi > 0} \xi + \frac{\theta_{t,(k)}^T \widetilde{D}_{t,(k)}(\xi)\theta_{t,(k)}}{\xi}. \tag{35}$$

When $\xi > 0$ we can differentiate and see that it satisfies (34). Since (35) is also convex, when (34) has a solution we know that this solution uniquely identifies the global minimizer $\xi$. The function $\sum_{i \in G_k} \left( \frac{[D_{t,(k)}]_{ii}[\theta_{t,(k)}]_i}{\xi[D_{t,(k)}]_{ii} + \alpha_t \lambda} \right)^2$ is monotonically decreasing in $\xi$. Hence (34) has a solution when the function at $\xi = 0$ is greater than 1, and this is equivalent to the condition in (33). $\square$

In conclusion, to compute the proximal mapping for each group, we check the condition in (33) and find $\xi$ using a root-finding method (e.g. bisection) observing that we have simple bounds for the root of (34): $0 < \xi < |G_k| \max_{i \in G_k}([D_{t,(k)}]_{ii})$, where $|G_k|$ is the cardinality of group $G_k$.

# G   Examples Satisfying Condition (C-4)

We provide concrete examples and derivations satisfying Condition (C-4) in Section 3. Before presenting our derivations, we need the following important theorem.

**Theorem 2** (Weyl). *For any two $n \times n$ Hermitian matrices $A$ and $B$, assume that the eigenvalues of $A$ and $B$ are*

$$\mu_1 \geq \cdots \geq \mu_n, \quad \text{and} \quad \nu_1 \geq \cdots \geq \nu_n$$

*respectively. Let $\lambda_1 \geq \cdots \geq \lambda_n$ be the eigenvalues of the matrix $A + B$, then the following holds*

$$\mu_j + \nu_k \leq \lambda_i \leq \mu_r + \nu_s$$

*for $j + k - n \geq i \geq r + s - 1$. Hence, we could derive*

$$\lambda_1 \leq \mu_1 + \nu_1$$

**Algorithm 2** PROXGENW: A **Gen**eral Stochastic **Prox**imal Gradient Method with Weight Decay

---

1: **Input:** Stepsize $\alpha_t$, $\{\rho_t\}_{t=1}^{t=T} \in [0,1)$, regularization parameter $\lambda$, small constant $0 < \delta \ll 1$, and weight decay regularization parameter $\zeta$.
2: **Initialize:** $\theta_1 \in \mathbb{R}^d$, $m_0 = 0$, and $C_0 = 0$.
3: **for** $t = 1, 2, \ldots, T$ **do**
4:     Draw a minibatch sample $\xi_t$ from $\mathbb{P}$
5:     $g_t \leftarrow \nabla f(\theta_t; \xi_t)$                    ▷ Stochastic gradient at time $t$
6:     $m_t \leftarrow \rho_t m_{t-1} + (1 - \rho_t) g_t$            ▷ First-order momentum estimate
7:     $C_t \leftarrow$ Preconditioner construction
8:     $\bar{\theta}_t \leftarrow (1 - \alpha_t \zeta)\theta_t$                    ▷ Apply decoupled weight decay
9:     $\theta_{t+1} \in \underset{\theta \in \Omega}{\arg\min} \left\{ \langle m_t, \theta \rangle + \lambda \mathcal{R}(\theta) + \frac{1}{2\alpha_t}(\theta - \bar{\theta}_t)^\mathsf{T}(C_t + \delta I)(\theta - \bar{\theta}_t) \right\}$
10: **end for**
11: **Output:** $\theta_T$

---

Now, we derive the $\gamma$ in (C-4) for various popular optimization algorithms used in deep learning communities.

**ADAGRAD.** In PROXGEN framework, ADAGRAD corresponds to $C_t = \left( \frac{1}{t} \sum_{\tau=1}^{t} g_\tau g_\tau^\mathsf{T} \right)^{1/2}$. Under the constant stepsizes $\alpha_t = \alpha$, we have

$$\lambda_{\max}(C_t) = \frac{1}{\sqrt{t}} \lambda_{\max} \left( \sum_{\tau=1}^{t} g_\tau g_\tau^\mathsf{T} \right)^{1/2}$$

$$\leq \frac{1}{\sqrt{t}} \left( \sum_{\tau=1}^{t} \lambda_{\max}(g_\tau g_\tau^\mathsf{T}) \right)^{1/2}$$

$$= \frac{1}{\sqrt{t}} \left( \sum_{\tau=1}^{t} \|g_\tau\|_2^2 \right)^{1/2}$$

$$\leq G$$

Hence, the Condition (C-4) can be satisfied as

$$\lambda_{\min}(\alpha_t(C_t + \delta I)^{-1}) \geq \frac{\alpha}{G + \delta} := \gamma$$

**RMSPROP and ADAM.** Exponential moving average (a.k.a. EMA) approaches correspond to $C_t = \left( \beta C_{t-1} + (1 - \beta) g_t g_t^\mathsf{T} \right)^{1/2}$ where $\beta \in [0,1)$ and $g_t$ denotes the stochastic gradient at time $t$. The usual RMSPROP and ADAM use diagonal approximations for $g_t g_t^\mathsf{T}$, but here we consider more general form (i.e. including general full matrix gradient outer-product) as introduce in [39]. First, we derive the upper bound for maximum eigenvalue for the matrix $C_t$. The matrix $C_t$ can be expressed by

$$C_t = \left( \beta C_{t-1} + (1 - \beta) g_t g_t^\mathsf{T} \right)^{1/2}$$

$$= \left( \beta^2 C_{t-2} + \beta(1 - \beta) g_{t-1} g_{t-1}^\mathsf{T} + (1 - \beta) g_t g_t^\mathsf{T} \right)^{1/2}$$

$$= \cdots$$

$$= \left( (1 - \beta) \sum_{i=1}^{t} \beta^{t-i} g_i g_i^\mathsf{T} \right)^{1/2}$$

We can derive the upper bound by

$$\lambda_{\max}(C_t) = \lambda_{\max} \left( (1 - \beta) \sum_{i=1}^{t} \beta^{t-i} g_i g_i^\mathsf{T} \right)^{1/2}$$

$$\leq \left( (1 - \beta) \sum_{i=1}^{t} \beta^{t-i} \lambda_{\max}(g_i g_i^\mathsf{T}) \right)^{1/2}$$

$$\leq \left( (1 - \beta) G^2 \sum_{i=1}^{t} \beta^{t-i} \right)^{1/2}$$

$$\leq G(1 - \beta^t)^{1/2} \leq G$$

Hence, we have $\lambda_{\max}(C_t + \delta I) \leq dG + \delta$. Also, we have

$$\lambda_{\max}(C_t + \delta I) = \frac{1}{\lambda_{\min}((C_t + \delta I)^{-1})} \leq \frac{1}{G + \delta}$$

Therefore, the condition (C-4) under the constant stepsize $\alpha_t = \alpha$ can be derived as

$$\lambda_{\min}(\alpha_t(C_t + \delta I)^{-1}) \geq \frac{\alpha}{G + \delta}$$

which yields $\gamma = \frac{\alpha}{G+\delta}$.

**Natural Gradient Descent.** In this case, we derive the condition (C-4) for the Fisher information matrix when the loss function is defined as a negative log-likelihood, i.e., $f = \log p(x|\theta)$. The natural gradient descent aims at considering general geometry (not limited to Euclidean geometry), but we restrict our focus on the distribution space where the Fisher information is employed for preconditioner matrix $C_t$. The Fisher information matrix is defined as

$$F = \mathbb{E}_{Q(x)P(y|x,\theta)}\left[\frac{\partial f(x|\theta)}{\partial \theta}\frac{\partial f(x|\theta)}{\partial \theta}^{\mathsf{T}}\right]$$

where $Q(x)$ is data distribution and $P(y|x,\theta)$ denotes the model's predictive distribution (ex. neural networks). However, in general, we do not have access to true data distribution, so we instead take an expectation with respect to empirical (training) data distribution $\widehat{Q}(x)$. This trick is also employed for K-FAC approximations to the Fisher [36]. Let the training samples be $\mathcal{S} = \{x_1, \cdots, x_n\}$ with sample size $n$. Then, the empirical Fisher could be computed as

$$\begin{aligned}
\widehat{F} &= \mathbb{E}_{\widehat{Q}(x)P(y|x,\theta)}\left[\frac{\partial f(x|\theta)}{\partial \theta}\frac{\partial f(x|\theta)}{\partial \theta}^{\mathsf{T}}\right] \\
&= \frac{1}{n}\sum_{i=1}^{n}\frac{\partial f(x_i|\theta)}{\partial \theta}\frac{\partial f(x_i|\theta)}{\partial \theta}^{\mathsf{T}}
\end{aligned}$$

Now, we bound the maximum eigenvalue of $\widehat{F}$ as

$$\begin{aligned}
\lambda_{\max}(\widehat{F}) &\leq \frac{1}{n}\sum_{i=1}^{n}\lambda_{\max}\left(\frac{\partial f(x_i|\theta)}{\partial \theta}\frac{\partial f(x_i|\theta)}{\partial \theta}^{\mathsf{T}}\right) \\
&\leq \frac{1}{n}\sum_{i=1}^{n}G^2 \\
&= G^2
\end{aligned}$$

by our Condition (C-3). Hence, the Condition (C-4) can be derived as

$$\lambda_{\min}(\alpha_t(\widehat{F} + \delta I)^{-1}) \geq \frac{\alpha}{G^2 + \delta}$$

under the constant stepsize $\alpha_t = \alpha$.

# H   Proofs of Theorem 1

**Lemma 2.** *The first-order momentum $m_t$ in Algorithm 1 satisfies*

$$\|m_t\|_2 \leq G$$

*Proof.* We use mathematical induction. For $t = 1$, the momentum is computed as $m_1 = \rho_1 m_0 + (1 - \rho_1)g_t = (1 - \rho_0)g_1$. Therefore, we have $\|m_t\|_2 = \|(1 - \rho_0)g_1\| \leq (1 - \rho_0)G \leq G$.

Now, we assume that $\|m_{t-1}\|_2 \leq G$ holds. The momentum at time $t$ is constructed by $m_t = (1 - \rho_t)m_{t-1} + \rho_t g_t$. Then, we have

$$\begin{aligned}
\|m_t\|_2 &= \|(1 - \rho_t)m_{t-1} + \rho_t g_t\|_2 \\
&\leq (1 - \rho_t)\|m_{t-1}\|_2 + \rho_t\|g_t\|_2 \\
&\leq (1 - \rho_t)G + \rho_t G = G
\end{aligned}$$

where the first inequality comes from the triangle inequality and the second one is derived from the induction hypothesis. □

We deal with the following update rule in Algorithm 1 as

$$\theta_{t+1} \in \operatorname*{argmin}_{\theta \in \Omega} \left\{ \left\langle (1 - \rho_t) g_t + \rho_t m_{t-1}, \theta \right\rangle + \mathcal{R}(\theta) + \frac{1}{2\alpha_t} (\theta - \theta_t)^\mathsf{T} (C_t + \delta I)(\theta - \theta_t) \right\} \qquad (36)$$

By the optimality condition, we have

$$0 \in (1 - \rho_t) g_t + \rho_t m_{t-1} + \widehat{\partial} \mathcal{R}(\theta_{t+1}) + \frac{1}{\alpha_t}(C_t + \delta I)(\theta_{t+1} - \theta_t)$$

which means that

$$-(1 - \rho_t) g_t - \rho_t m_{t-1} - \frac{1}{\alpha_t}(C_t + \delta I)(\theta_{t+1} - \theta_t) \in \widehat{\partial} \mathcal{R}(\theta_{t+1})$$

By adding the gradient $\nabla f(\theta_{t+1})$ on both sides, we have

$$\nabla f(\theta_{t+1}) - (1 - \rho_t) g_t - \rho_t m_{t-1} - \frac{1}{\alpha_t}(C_t + \delta I)(\theta_{t+1} - \theta_t) \in \nabla f(\theta_{t+1}) + \widehat{\partial} \mathcal{R}(\theta_{t+1}) = \widehat{\partial} F(\theta_{t+1})$$

By the definition of $\theta_{t+1}$ in (36), we obtain

$$\left\langle (1 - \rho_t) g_t + \rho_t m_{t-1}, \theta_{t+1} \right\rangle + \mathcal{R}(\theta_{t+1}) + \frac{1}{2\alpha_t}(\theta_{t+1} - \theta_t)^\mathsf{T}(C_t + \delta I)(\theta_{t+1} - \theta_t)$$

$$\leq \left\langle (1 - \rho_t) g_t + \rho_t m_{t-1}, \theta_t \right\rangle + \mathcal{R}(\theta_t)$$

which in result

$$\left\langle (1 - \rho_t) g_t + \rho_t m_{t-1}, \theta_{t+1} - \theta_t \right\rangle + \mathcal{R}(\theta_{t+1}) + \frac{1}{2\alpha_t}(\theta_{t+1} - \theta_t)^\mathsf{T}(C_t + \delta I)(\theta_{t+1} - \theta_t) \leq \mathcal{R}(\theta_t)$$

Since the function $f$ is $L$-smooth by Condition (C-1), we have

$$f(\theta_{t+1}) \leq f(\theta_t) + \left\langle \nabla f(\theta_t), \theta_{t+1} - \theta_t \right\rangle + \frac{L}{2}\|\theta_{t+1} - \theta_t\|_2^2$$

Adding previous two inequalities yields

$$\left\langle (1 - \rho_t) g_t - \nabla f(\theta_t) + \rho_t m_{t-1}, \theta_{t+1} - \theta_t \right\rangle + (\theta_{t+1} - \theta_t)^\mathsf{T} \left( \frac{1}{2\alpha_t}(C_t + \delta I) - \frac{L}{2}I \right)(\theta_{t+1} - \theta_t)$$

$$\leq F(\theta_t) - F(\theta_{t+1}) \qquad (37)$$

Then, we have

$$\|\theta_{t+1} - \theta_t\|^2_{\frac{1}{2\alpha_t}(C_t + \delta I) - \frac{L}{2}I}$$

$$\overset{\textcircled{1}}{\leq} F(\theta_t) - F(\theta_{t+1}) - \left\langle (1 - \rho_t) g_t - \nabla f(\theta_t), \theta_{t+1} - \theta_t \right\rangle - \left\langle \rho_t m_{t-1}, \theta_{t+1} - \theta_t \right\rangle$$

$$= F(\theta_t) - F(\theta_{t+1}) - \left\langle g_t - \nabla f(\theta_t), \theta_{t+1} - \theta_t \right\rangle + \left\langle \rho_t g_t, \theta_{t+1} - \theta_t \right\rangle - \left\langle \rho_t m_{t-1}, \theta_{t+1} - \theta_t \right\rangle$$

$$\overset{\textcircled{2}}{\leq} F(\theta_t) - F(\theta_{t+1}) + \frac{1}{2L}\|g_t - \nabla f(\theta_t)\|_2^2 + \frac{L}{2}\|\theta_{t+1} - \theta_t\|_2^2 + \frac{\rho_t^2}{2L}\|g_t\|_2^2 + \frac{L}{2}\|\theta_{t+1} - \theta_t\|_2^2$$

$$\quad + \|\rho_t m_{t-1}\|_2 \|\theta_{t+1} - \theta_t\|_2$$

$$\overset{\textcircled{3}}{\leq} F(\theta_t) - F(\theta_{t+1}) + \rho_0 \mu^{t-1} D G + \frac{\rho_0^2 \mu^{2(t-1)} G^2}{2L} + L\|\theta_{t+1} - \theta_t\|_2^2 + \frac{1}{2L}\|g_t - \nabla f(\theta_t)\|_2^2$$

The derivations in inequalities (1-3) as follows:

   ①  We rearrange the inequality (37).

   ②  We use the fact that $\langle a, b \rangle \leq \frac{1}{2}\|a\|_2^2 + \frac{1}{2}\|b\|_2^2$ and $\langle a, b \rangle \leq \|a\|_2 \|b\|_2$. With this, we use modified version such as $\langle a, b \rangle = \langle ca, \frac{1}{c}b \rangle \leq c^2 \|a\|_2^2 + \frac{1}{c^2}\|b\|_2^2$ for any positive constant $c$.

   ③  We apply our Lemma 2 and Condition (C-3).

By rearranging the above inequality, we require the following quantity be positive-semidefinite.

$$\frac{1}{2\alpha_t}(C_t + \delta I) - \frac{3}{2}LI \succeq 0$$

Note that in this inequality we can see that

$$\frac{1}{2\alpha_t}(C_t + \delta I) - \frac{3}{2}LI \succeq \frac{1}{2\alpha_0}\delta I - \frac{3}{2}LI$$

since $C_t$ is positive (semi)definite and $\alpha_t$ is *non-increasing*. Therefore, from this we can derive the stepsize condition in our Theorem 1 as

$$\alpha_0 \leq \frac{\delta}{3L}$$

Therefore, we have

$$\sum_{t=0}^{T-1} \|\theta_{t+1} - \theta_t\|^2_{\frac{1}{2\alpha_t}(C_t + \delta I) - \frac{3}{2}LI} \leq \underbrace{F(\theta_0) - F(\theta^*)}_{\Delta} + \underbrace{\frac{\rho_0 DG}{1 - \mu} + \frac{\rho_0^2 G^2}{2L(1 - \mu^2)}}_{C_1} + \frac{1}{2L} \sum_{t=0}^{T-1} \|g_t - \nabla f(\theta_t)\|^2_2$$

$$\leq \Delta + C_1 + \frac{1}{2L} \sum_{t=0}^{T-1} \|g_t - \nabla f(\theta_t)\|^2_2$$

Furthermore, we also have by stepsize condition

$$\left(\frac{\delta}{2\alpha_0} - \frac{3}{2}L\right) \sum_{t=0}^{T-1} \|\theta_{t+1} - \theta_t\|^2_2 \leq \sum_{t=0}^{T-1} \|\theta_{t+1} - \theta_t\|^2_{\frac{1}{2\alpha_t}(C_t + \delta I) - \frac{3}{2}LI} \leq \Delta + C_1 + \frac{1}{2L} \sum_{t=0}^{T-1} \|g_t - \nabla f(\theta_t)\|^2_2$$

since $\delta I \preceq C_t + \delta I$. From above inequality, we obtain

$$\sum_{t=0}^{T-1} \|\theta_{t+1} - \theta_t\|^2_2 \leq H_1 + H_2 \sum_{t=0}^{T-1} \|g_t - \nabla f(\theta_t)\|^2_2 \tag{38}$$

where the constants $H_1$ and $H_2$ are defined as

$$H_1 = \Delta \Big/ \left(\frac{\delta}{2\alpha_0} - \frac{3}{2}L\right) + C_1 \Big/ \left(\frac{\delta}{2\alpha_0} - \frac{3}{2}L\right)$$

$$H_2 = \frac{1}{2L\left(\frac{\delta}{2\alpha_0} - \frac{3}{2}L\right)}$$

Our goal is to bound the distance between the zero vector and subdifferential set of $F$, so we have

$$\text{dist}(\mathbf{0}, \widehat{\partial} F(\theta_{t+1}))^2$$

$$= \left\| (1 - \rho_t)g_t - \nabla f(\theta_{t+1}) + \rho_t m_{t-1} + \frac{1}{\alpha_t}(C_t + \delta I)(\theta_{t+1} - \theta_t) \right\|^2_2$$

$$= \left\| (1 - \rho_t)g_t - \nabla f(\theta_{t+1}) + \rho_t m_{t-1} + (\theta_{t+1} - \theta_t) + \frac{1}{\alpha_t}(C_t + \delta I)(\theta_{t+1} - \theta_t) - (\theta_{t+1} - \theta_t) \right\|^2_2$$

$$\leq 3 \left\| (1 - \rho_t)g_t - \nabla f(\theta_{t+1}) + \rho_t m_{t-1} + (\theta_{t+1} - \theta_t) \right\|^2_2$$

$$+ 3 \left\| \frac{1}{\alpha_t}(C_t + \delta I)(\theta_{t+1} - \theta_t) \right\|^2_2 + 3 \left\| (\theta_{t+1} - \theta_t) \right\|^2_2$$

$$\leq 3 \underbrace{\left\| (1 - \rho_t)g_t - \nabla f(\theta_{t+1}) + \rho_t m_{t-1} + (\theta_{t+1} - \theta_t) \right\|^2_2}_{T_1} + 3\left(\frac{1}{\gamma^2} + 1\right) \|\theta_{t+1} - \theta_t\|^2_2$$

Here, we assume that

$$\lambda_{\max}\left(\frac{1}{\alpha_t}(C_t + \delta I)\right) \leq \frac{1}{\gamma}$$

which yields our Condition (C-4)

$$\lambda_{\min}\left(\alpha_t(C_t + \delta I)^{-1}\right) \geq \gamma$$

From (37), we have

$$\left\langle (1 - \rho_t)g_t - \nabla f(\theta_t) + \rho_t m_{t-1}, \theta_{t+1} - \theta_t \right\rangle + \|\theta_{t+1} - \theta_t\|^2_{\frac{1}{2\alpha_t}(C_t + \delta I) - \frac{L}{2}I} \leq F(\theta_t) - F(\theta_{t+1})$$

which can be re-written as

$$\left\langle (1 - \rho_t)g_t - \nabla f(\theta_{t+1}) + \rho_t m_{t-1}, \theta_{t+1} - \theta_t \right\rangle$$

$$\leq F(\theta_t) - F(\theta_{t+1}) - \left\langle \nabla f(\theta_{t+1}) - \nabla f(\theta_t), \theta_{t+1} - \theta_t \right\rangle - \|\theta_{t+1} - \theta_t\|^2_{\frac{1}{2\alpha_t}(C_t + \delta I) - \frac{L}{2}I}$$

$$\leq F(\theta_t) - F(\theta_{t+1}) - \left\langle \nabla f(\theta_{t+1}) - \nabla f(\theta_t), \theta_{t+1} - \theta_t \right\rangle + \left(\frac{\delta}{2\alpha_0} - \frac{L}{2}\right) \|\theta_{t+1} - \theta_t\|^2_2$$

since we have the condition $\frac{\delta}{2\alpha_0} \geq \frac{3}{2}L$. Therefore, we obtain

$$
\begin{aligned}
T_1 &= \|(1-\rho_t)g_t - \nabla f(\theta_{t+1}) + \rho_t m_{t-1}\|_2^2 + \|\theta_{t+1} - \theta_t\|_2^2 \\
&\quad + 2\Big\langle (1-\rho_t)g_t - \nabla f(\theta_{t+1}) + \rho_t m_{t-1}, \theta_{t+1} - \theta_t \Big\rangle \\
&\leq \|(1-\rho_t)g_t - \nabla f(\theta_t) + \nabla f(\theta_t) - \nabla f(\theta_{t+1}) + \rho_t m_{t-1}\|_2^2 + \|\theta_{t+1} - \theta_t\|_2^2 \\
&\quad + F(\theta_t) - F(\theta_{t+1}) - \big\langle \nabla f(\theta_{t+1}) - \nabla f(\theta_t), \theta_{t+1} - \theta_t \big\rangle + \Big(\frac{\delta}{2\alpha_0} - \frac{L}{2}\Big)\|\theta_{t+1} - \theta_t\|^2 \\
&\leq 4\|g_t - \nabla f(\theta_t)\|_2^2 + 4L^2\|\theta_{t+1} - \theta_t\|_2^2 + 4\|\rho_t m_{t-1}\|_2^2 + 4\|\rho_t g_t\|_2^2 + \|\theta_{t+1} - \theta_t\|_2^2 \\
&\quad + F(\theta_t) - F(\theta_{t+1}) + L\|\theta_{t+1} - \theta_t\|_2^2 + \Big(\frac{\delta}{2\alpha_0} - \frac{L}{2}\Big)\|\theta_{t+1} - \theta_t\|_2^2 \\
&\leq F(\theta_t) - F(\theta_{t+1}) + 4\rho_0^2 \mu^{2(t-1)}G^2 + 4\rho_0^2 \mu^{2(t-1)}G^2 \\
&\quad + \Big(\frac{\delta}{2\alpha_0} + \frac{L}{2} + 1 + 4L^2\Big)\|\theta_{t+1} - \theta_t\|_2^2 + 4\|g_t - \nabla f(\theta_t)\|_2^2
\end{aligned}
$$

Therefore, we have the distance as

$$
\begin{aligned}
&\operatorname{dist}\big(\mathbf{0}, \widehat{\partial}F(\theta_{t+1})\big)^2 \\
&\leq 3\Bigg( F(\theta_t) - F(\theta_{t+1}) + 8\rho_0^2 \mu^{2(t-1)}G^2 + \underbrace{\Big(\frac{\delta}{2\alpha_0} + \frac{L}{2} + 2 + 4L^2 + \frac{1}{\gamma^2}\Big)}_{C_2}\|\theta_{t+1} - \theta_t\|_2^2 + 4\|g_t - \nabla f(\theta_t)\|_2^2 \Bigg)
\end{aligned}
$$

Therefore, we have

$$
\begin{aligned}
\mathbb{E}[\operatorname{dist}\big(\mathbf{0}, \widehat{\partial}F(\theta_a)\big)^2] &\leq \frac{1}{T}\sum_{t=0}^{T-1}\mathbb{E}\Big[\big\|(1-\rho_t)g_t - \nabla f(\theta_{t+1}) + \rho_t m_{t-1} + \frac{1}{\alpha_t}(C_t + \delta I)(\theta_{t+1} - \theta_t)\big\|_2^2\Big] \\
&\leq \frac{3}{T}\Big(\Delta + \frac{8\rho_0^2 G^2}{1-\mu^2} + 4\sum_{t=0}^{T-1}\|g_t - \nabla f(\theta_t)\|_2^2 + C_2 \sum_{t=0}^{T-1}\|\theta_{t+1} - \theta_t\|_2^2\Big) \\
&\leq \frac{3}{T}\Big(\Delta + \frac{8\rho_0^2 G^2}{1-\mu^2} + 4\sum_{t=0}^{T-1}\|g_t - \nabla f(\theta_t)\|_2^2 + C_2(H_1 + H_2\sum_{t=0}^{T-1}\|g_t - \nabla f(\theta_t)\|_2^2)\Big) \\
&\leq \frac{Q_1}{T}\sum_{t=0}^{T-1}\mathbb{E}\big[\|g_t - \nabla f(\theta_t)\|_2^2\big] + \frac{Q_2\Delta}{T} + \frac{Q_3}{T}
\end{aligned}
$$

where

$$
Q_1 = 4 + C_2 H_2, \quad Q_2 = 3 + \frac{3C_2}{\frac{\delta}{2\alpha_0} - \frac{3}{2}L}, \quad Q_3 = \frac{24\rho_0^2 G^2}{1-\mu^2} + \frac{3C_1 C_2}{\frac{\delta}{2\alpha_0} - \frac{3}{2}L}
$$

Note that the constants $Q_1$, $Q_2$, and $Q_3$ depend on $\{\alpha_0, \delta, L, D, G, \rho_0, \mu, \gamma\}$, but not on $T$. The third inequality comes from (38). If we assume the stochastic gradient $g_t$ is evaluated on the minibatch $\mathcal{S}_t$ with $|\mathcal{S}_t| = b_t$, then we can obtain using Condition (C-2)

$$
\begin{aligned}
\|g_t - \nabla f(\theta_t)\|_2^2 &= \mathbb{E}_\xi\Big[\big\|\frac{1}{b_t}\sum_{i=1}^{b_t}\nabla f(\theta_t; \xi_{i_t}) - \nabla f(\theta_t)\big\|_2^2\Big] \\
&= \frac{1}{b_t^2}\mathbb{E}\Big[\|\sum_{i=1}^{b_t}\{\nabla f(\theta_t; \xi_{i_t}) - \nabla f(\theta_t)\}\|_2^2\Big] \\
&\leq \frac{1}{b_t^2}\sum_{i_t=1}^{b_t}\mathbb{E}\big[\|\nabla f(\theta_t; \xi_{i_t}) - \nabla f(\theta_t)\|_2^2\big] \leq \frac{1}{b_t}\sigma^2
\end{aligned}
$$

where $i_t$ represents the random variable for each datapoint in minibatch samples $\mathcal{S}_t$. Finally, we arrive at our Theorem 1 as

$$
\mathbb{E}_R[\operatorname{dist}\big(\mathbf{0}, \widehat{\partial}F(\theta_R)\big)^2] \leq \frac{Q_1 \sigma^2}{T}\sum_{t=0}^{T-1}\frac{1}{b_t} + \frac{Q_2\Delta}{T} + \frac{Q_3}{T}
$$

It can be clearly seen from the definitions of $C_1, C_2, H_1$, and $H_2$ that the constants $\{Q_i\}_{i=1}^3$ in Theorem 1 absolutely do not involve the problem dimension $d$.