# OpenReview forum: "Adaptive Proximal Gradient Methods for Structured Neural Networks"
_NeurIPS.cc/2021/Conference — NeurIPS 2021 Poster_

### Official Review · Reviewer_REN2 · 2021-07-12

**Rating:** 6
**Confidence:** 3

**Summary:**

The paper presents a framework for adaptive proximal stochastic methods. The general algorithm provided, ProxGen, is a variation on traditional proximal gradient descent in which the gradient is multiplied by the inverse of some $C_t+ \delta I$, and the norm in the prox operator is given by $\|x\| = x^{\top} (C_t+ \delta I) x$. They give a convergence result which matches the optimal rate for SGD, and they show experiments with sparse, group-sparse, and binary neural networks in which they show improvements over existing algorithms.

**Limitations And Societal Impact:**

The limitations are discussed. There is no potential negative societal impact.

**Main Review:**

The algorithm presented is not that novel in itself, as it is very close to Prox-Quant (differing only in the norm used for the prox-operator), and to Prox-SGD (differing on the fact that Prox-SGD is a two step algorithm, so it seems that ProxGen is a “simplification” of Prox-SGD). Despite that, the authors generally do a good job in arguing the empirical and theoretical advantages of the proposed method w.r.t. the existing two.

Question: In lines 93-94, there is the statement that Adagrad is an instance of ProxGen with a certain choice of $C_t$ and $\mathcal{R}(\theta) = \|\theta\|_1$. Is that accurate? To my understanding, “vanilla” Adagrad does not include any proximal step. Should it say “proximal” Adagrad instead?

As to the proof of Theorem 1, is the proof structure based on any previous work, or is it completely new? I think it would be useful to give some pointers.
Although the steps to get Corollary 1 from Theorem 1 are not too involved, they are not a one-liner, and should be included either in the main text or the appendix. Also, add citation for the $O(1/\epsilon^4)$ optimal rate: this is relevant.

Regarding the experiments, the plots in Figure 2 should be uniformed: Why is Prox-SGD only in some plots (it does not look good)? Some plots have legends, some do not, and the legend for the last one is cryptic.

Typos: The paper needs to be proofread for typos. Line 64: “arbitrat”. Line 72: “approacwstygh”.

**Time Spent Reviewing:**

2.5

---

> ### Author Response · Authors · 2021-08-10
> **Response to Reviewer REN2**
>
> Dear Reviewer REN2,
>
> Thank you for your constructive feedback.
>
> -------------------------------------------
>
> [Q1: On AdaGrad]
>
> Since the original AdaGrad paper [A] designed "adaptive proximal function'', we implicitly assumed that AdaGrad contains a proximal step in its update rule. However, as the reviewer said, we will modify the term "proximal version of AdaGrad'' for clarity. Thank you for the suggestion.
>
> -------------------------------------------
>
> [Q2: On the proof of Theorem 1]
>
> The proof structure of our Theorem 1 is based on [B], but the detailed derivations are completely different from that of [B]. While [B] only considers the proximal gradient descent of vanilla SGD, ProxGen should handle the first-order momentum $m_t$ with arbitrary preconditioner $C_t$. Toward this, our analysis requires mild conditions (C-3) and (C-4) and successfully guarantees the optimal convergence rate $\mathcal{O}(1/\epsilon^4)$ under suitable batch size condition as we will show below.
>
> -------------------------------------------
>
> [Q3: On the proof of Corollary 1]
>
> Thank you for the suggestion. We provide the proof for Corollary 1 here. The batch size condition $b_t = b = \Theta(\sqrt{n})$ is the same as $b = \Theta(T)$ as follows. If we consider the $e$ epochs, then
> \begin{align*}
>     T = \frac{e \times n}{b} = \Theta(e \times \sqrt{n}) = \Theta(\sqrt{n}).
> \end{align*}
> Here we have $b = \Theta(T)$, so the first term in the right-hand side in our Theorem 1 becomes
> \begin{align*}
>     \frac{Q_1 \sigma^2}{T} \sum\limits_{t=0}^{T-1} \frac{1}{b_t} = \mathcal{O}(\frac{1}{T})
> \end{align*}
> which yields $\mathbb{E}_a\big[\mathrm{dist}(0, \widehat{\partial}F(\theta_a))^2\big] \le \mathcal{O}(1/T)$.
>
> -------------------------------------------
>
> [Q4: On the optimal rate]
>
> Thank you for pointing this out. By the work [C,D] on the lower bounds of finding stationary points, the optimal rate is known to be $\mathcal{O}(1/\epsilon^4)$. We will include these references in the revision.
>
> -------------------------------------------
>
> [Q5: On Figure 2]
>
> We apologize for the confusion. We did not maliciously select baselines according to experimental results. Rather, we used appropriate (but possibly different) baselines for each regularizer. Specifically, Prox-SGD [E] only targets "convex regularizer'' in its theory and experiments, so we included the results of Prox-SGD only for the $\ell_1$ regularizer in Figure 2-(a)  (We could also confirm that Prox-SGD can be applied to non-convex regularizers by force, but its performance is not good). Also, in Figure 2-(d), the $\ell_0$-regularized problem cannot be solved via subgradient-based methods, so we considered the approximated $\ell_0$-norm using hard concrete distribution [F] as a natural alternative baseline. We will clarify this in the revision. We will also unify the formats of figures. We would like to thank you again for your careful feedback.
>
> -------------------------------------------
>
> [Q6: On typos]
>
> Thank you for the correction. We will fix all typos in the revision.
>
> -------------------------------------------
>
> [References]
>
> [A] Adaptive Subgradient Methods for Online Learning and Stochastic Optimization, Duchi et al., JMLR 2011.
>
> [B] Non-asymptotic Analysis of Stochastic Problems for Non-Smooth Non-Convex Regularized Problems, Xu et al., NeurIPS 2019.
>
> [C] Lower Bounds for Non-convex Stochastic Optimization, Arjevani et al., 2019.
>
> [D] Lower Bounds for Finding Stationary Points, Carmon et al., 2017.
>
> [E] Prox-SGD: Training Structured Neural Networks under Regularization and Constraints, Yang et al., ICLR 2020.
>
> [F] Learning Sparse Neural Networks through $\ell_0$ Regularization, Louizos et al., ICLR 2018.

---

### Official Review · Reviewer_Uvqk · 2021-07-13

**Rating:** 7
**Confidence:** 4

**Summary:**

This work introduces a proximal algorithm for training neural networks. Specifically, the method allows the use of non-convex regularization functions and general PSD pre-conditioners  in the objective, while guaranteeing convergence to a stationary point. Empirical results are provided for training sparse neural networks on CIFAR and ImageNet.

**Limitations And Societal Impact:**

No need for societal impact.

**Main Review:**

### Summary
This is a well-motivated and well-executed work that demonstrates the effectiveness of its proposed proximal algorithm when employing  non-convex regularizers to train sparse or quantized neural networks.

### Strengths
1. This work is well-motivated, and provides an interesting method especially in the context of training sparse or quantized neural networks.
2. The convergence analysis provided in Theorem 1 seems to provide powerful results. The proof looks correct to me, although I have not checked all details.
3. The assumptions made for the theoretical results are diligently checked in practice through dedicated experiments.
4. The submission conducts convincing experiments that demonstrate the usefulness of the approach, including on the large-scale task of training a binary ResNet-18 on ImageNet.

### (Minor) Weaknesses
1. The paper is not easy to read and follow, perhaps due to its high density throughout.
2. Experiments are "only" conducted for training quantized / sparse neural networks, which are interesting topics by themselves, but perhaps special cases only compared to what the reader might expect from the generality of the method suggested by the title and abstract.

### Comments
1. In the proof of Theorem 1, L.777 should be an inequality rather than an equality (distance from set to 0 is $\leq$ norm of one element of the set).
2. Can the authors detail why the method converges with constant mini-batch size? It is not obvious how the the first term of the RHS (from Theorem 1) converges to zero when the batch-size is constant?

### Typos
1. L.44 Strategic -> strategy
2. l.64: arbitrat -> arbitrary
3. l.72 approachstygh
4. in caption of Table 4, “first” misspelled

----

### Updated Review

Thanks for the authors for engaging in a detailed discussion. It is slightly disappointing that Corollary 1 ends up requiring the assumption $b = \Theta(T)$ since it is less elegant than the initial assumption $b = \Theta(\sqrt{n})$, but that is not a major issue.

I am keeping my initial rating of 7. Whether the submission gets accepted or is resubmitted to other venues, I would encourage the authors to take the time and effort to improve clarity of the submission (potentially making space for expanding on the most important points by deferring lower level-details to the appendix).

**Time Spent Reviewing:**

6

---

> ### Author Response · Authors · 2021-08-10
> **Response to Reviewer Uvqk**
>
> Dear Reviewer Uvqk,
>
> Thank you for your valuable feedback.
>
> ------------------------------------------------------------
>
> [Q1: On experiments.]
>
> To illustrate the generality of our method, we also included experiments on the task of detecting non-linear Granger Causality in multivariate time-series data in Appendix D. For this task, [A] considers the proximal update of Adam, but [A] does not take into account the preconditioner in the proximal operator as in ProxQuant [B] (see also line 132, paragraph "Revised ProxQuant'' in Section 2 of our paper). Table 6 in Appendix D shows the superiority of ProxGen over the baseline [A] on several time-series datasets including  Lorenz, VAR, and Dream-3 [A] for Granger causal inference.
>
> To further showcase the usefulness of our framework in deep learning tasks, we also plan to conduct simple experiments for continual learning. We will incorporate the results in the revision.
>
> ------------------------------------------------------------
>
> [Q2: On the proof of Theorem 1]
>
> Thank you for the correction. As noted by the reviewer, the equality at line 777 in Appendix should be an inequality. We will fix it in the revision.
>
> ------------------------------------------------------------
>
> [Q3: On the proof of Corollary 1]
>
> Thank you for the suggestion. We provide the proof for Corollary 1 here. The batch size condition $b_t = b = \Theta(\sqrt{n})$ is the same as $b = \Theta(T)$ as follows. If we consider the $e$ epochs, then
> \begin{align*}
>     T = \frac{e \times n}{b} = \Theta(e \times \sqrt{n}) = \Theta(\sqrt{n}).
> \end{align*}
> Here we have $b = \Theta(T)$, so the first term in the right-hand side in our Theorem 1 becomes
> \begin{align*}
>     \frac{Q_1 \sigma^2}{T} \sum\limits_{t=0}^{T-1} \frac{1}{b_t} = \mathcal{O}(\frac{1}{T})
> \end{align*}
> which yields $\mathbb{E}_a\big[\mathrm{dist}(0, \widehat{\partial}F(\theta_a))^2\big] \le \mathcal{O}(1/T)$.
>
> ------------------------------------------------------------
>
> [Q4: On typos]
>
> Thank you for the correction. We will fix all typos in the revision.
>
> ------------------------------------------------------------
>
> In the revised version, we will seek to improve the clarity and readability of the presentation.
>
> ------------------------------------------------------------
>
> [References]
>
> [A] Economy Statistical Recurrent Units for Inferring Nonlinear Granger Causality, Khanna et al., ICLR 2020.
>
> [B] ProxQuant: Quantized Neural Networks via Proximal Operators, Bai et al., ICLR 2019.
>
> [C] Overcoming Catastrophic Forgetting for Continual Learning via Model Adaptation, Hu et al., ICLR 2019.

---

> > ### Comment · Reviewer_Uvqk · 2021-08-23
> > **Re Q3: On the proof of Corollary 1**
> >
> > Thank you for the detailed response.
> >
> > The provided proof for Corollary 1 does not look rigorous enough, because the number of epochs $e$ is not a fixed number (it depends on $n$ and $T$).
> >
> > At a higher level, it does not seem possible that assuming $b_t = \Theta(\sqrt{n})$ always gives $b_t = \Theta(T)$, since the dataset size $n$ and the number of iterations $T$ are two independent variables in general -- unless $T$ is explicitly made dependent on $n$, which does not seem to be the case here?

---

> > > ### Author Response · Authors · 2021-08-25
> > > **Response to Additional Comments**
> > >
> > > Thank you for the additional comments.
> > >
> > > As the reviewer noted, the sample size $n$ and the number of iterations $T$ can be set independently in general. In terms of stochastic optimization, however, it is more natural in practice to choose the batch size $b$ and the number of epochs $e$ (rather than the number of iterations) in advance. Then the total number of iterations $T$ is determined according to $b$ and $e$: $T=e \times n/b$. In this sense, for the batch size $b_t = \Theta(\sqrt{n})$ and the number of epochs $e$, the total iterations $T$ should be an order of $e \times n / \Theta(\sqrt{n}) = \Theta(\sqrt{n})$. In summary, the number of epochs $e$ and the sample size $n$ are independent to each other, and the total iterations $T$ is determined by $n$ and $e$.

---

> > > > ### Comment · Reviewer_Uvqk · 2021-08-26
> > > > **Response**
> > > >
> > > > Thanks for the further clarification. Where I am getting at is that it can be misleading to have $e$ disappear as a constant inside the $\Theta$ notation because it still holds a dependency on $T$ and $n$.
> > > >
> > > > More specifically, the term that we are discussing is:
> > > > $$\dfrac{Q_1 \sigma^2}{T} \sum\limits_{t=0}^{T-1} \dfrac{1}{b_t} = \dfrac{Q_1 \sigma^2}{b}$$
> > > >
> > > > We can choose the value of the batch-size $b$ but it cannot be larger than the dataset $n$, thus at best, this term cannot be smaller than $\dfrac{Q_1 \sigma^2}{n}$, which does not depend on the number of iterations or epochs. Therefore it is unclear to me how, as stated in Corollary 1, this term can converge to $0$ as $T \to \infty$ without further assumptions.

---

> > > > > ### Author Response · Authors · 2021-08-29
> > > > > **Response to Additional Comments**
> > > > >
> > > > > Thank you for your additional comments.
> > > > >
> > > > > For theoretical completeness, we will add to our batch size condition the standard assumption $b_t = \Theta(T)$ as already considered in many previous studies such as [A, B, C]. Then, the first term in the right-hand side of our Theorem 1 clearly yields $\mathcal{O}(1/T)$. We will also include the discussion on $b_t = \Theta(\sqrt{n})$ as a remark. Thank you again for your valuable suggestion.
> > > > >
> > > > > [A] Adaptive Methods for Nonconvex Optimization, Zaheer et al., NeurIPS 2018.
> > > > >
> > > > > [B] Non-asymptotic Analysis of Stochastic Methods for Non-smooth Non-convex Regularized Problems, Xu et al., NeurIPS 2019.
> > > > >
> > > > > [C] Structured Sparsity Inducing Adaptive Optimizers for Deep Learning, T. Deleu et al., 2021.

---

### Official Review · Reviewer_3MNN · 2021-07-15

**Rating:** 5
**Confidence:** 3

**Summary:**

This paper proposes adaptive proximal gradient methods for nonconvex optimization where the regularizer can be nonsmooth and nonconvex. The paper also provides extensive experiments to show that the proposed approach outperform existing SGD methods for stochastic optimization.

**Limitations And Societal Impact:**

The choice of stepsize is unclear.

**Main Review:**

I feel that the theoretical contribution as well as the novelty of this paper is quite limited, and this paper did not broaden our understanding of adaptive methods for sparse optimization. The main weakness of this paper is that the convergence analysis is not technically sound. A potential limitation is that the stepsize of the proposed algorithm  virtually has nothing to do with the adaptive metric. In fact, by using some nonstandard assumption C3, C4, the convergence analysis is nearly identical to the standard analysis of stochastic PGD.

Furthermore, I feel that some of the theoretical results are incorrect or at least have obvious limitations.
- The stepsize is the main tuning parameter for ProxGen.  A close look at the convergence analysis reveals that the tuning parameter has nothing to do with the adaptive metric at all. It is somewhat strange that the stepsize solely depends on the free parameter $\delta$ which is only a small to term to ensure the posititive definiteness of the metric. So how should the algorithm choose $\delta$?
- Line 236 the paper claims the advantage of ProxGen. However, this bound on $1/\gamma$ is different from the bound derived in appendix line 733. Is there a typo? It is also bothering that  $1/\gamma$ depends on $1/\delta$, which can be arbitrarily small, this goes back to the previous question on how to select $ delta$.
- This paper considers $\ell_0$ penalty as a motivated example. However, it seems inappropriate for the theory. Specifically, Definition 1 (Frechet subdifferential) is not well-defined for the proposed $\ell_0$ regularizer. Taking $R(\theta)=\||\theta\||_0$ in the 1-d case, if $\theta=0$, then  $\partial R(0)=\[\theta^*, |\liminf  \frac{1-\langle \theta^*, x-0\rangle}{|x|} \ge 0\]=\mathbb{R}$. Therefore, the subdifferential of the objective always contains zero.
- The author proposes C3 in which the difference between iterates is bounded by $D$. This appears to be unnecessary. In the proof (line 763) one can subtract $(L+\rho_0G)\||\theta_{t+1}-\theta_{t}\||^2$ on both sides. Then using quadratic bound, one can easily remove the term $\||\theta_{t+1}-\theta_{t}\||$. It suffices to impose a slightly tighter bound on $\alpha_0$.

**Time Spent Reviewing:**

6

---

> ### Author Response · Authors · 2021-08-09
> **Response to Reviewer 3MNN**
>
> Dear Reviewer 3MNN,
>
> Thank you for your constructive feedback
>
> ---------------------------------------------------------------------------------------------------
>
> [Q1: On stepsize]
>
> Contrary to the reviewer's belief, the stepsize is naturally formulated as being independent of the adaptive metric in our ProxGen framework and the adaptivity is solely determined by the preconditioning matrix $C_t$ as considered in **all** previous works of Table 1 and [3,4,5,6,7] (note that the learning rate, which is scalar, cannot realize different/adaptive learning rates for each coordinate). Especially, the stepsize condition $\alpha_0 \le \delta/cL$ (for some positive constant $c$, and in our case, $c = 3$) is very standard in the literature of adaptive gradient methods [3,4,5,6,7]. Since $\delta$ is a free parameter, one can set arbitrary value for $\delta$ in theory and then initial learning rate $\alpha_0$ could be determined by the stepsize condition. After the initial stepsize $\alpha_0$ is fixed, any non-increasing learning rate scheduling for $\alpha_t$ is allowed and the convergence is guaranteed for such learning rate scheduling due to our Theorem 1.
>
> In conclusion, considering the learning rate independent of the adaptive metric is a natural formulation and cannot be a weakness of our analysis. Rather, our convergence analysis successfully  reflects the advantages of using adaptive gradient methods (see line 230) and is **never** nearly identical to the existing analysis for stochastic PGD (see line 192 for challenges in considering the general forms of adaptivity).
>
> ---------------------------------------------------------------------------------------------------
>
> [Q2: On $1/\gamma$]
>
> This is another misunderstanding of the reviewer since the quantity $1/\gamma$ does not simply "depend on $1/\delta$'' but rather is proportional to $\delta$ as in our derivation in Appendix line 733. In line $236$ regarding $1/\gamma$ for Adam's update rule, we omit the effect of $\delta$ to clearly see the advantages of adaptive gradient methods. However, if we consider $\delta$ here, we can obtain exactly the same results as in Appendix line 733 (at Appendix line 733, $\gamma \ge \frac{\alpha}{dG+\delta}$ should be $\frac{\alpha}{G+\delta}$. We apologize for this typo) as follows
>
> \begin{align}
>     Q_i \propto 1/\gamma = \frac{\sqrt{(1 - \beta) \sum\limits_{\tau=1}^{t}\beta^{t-\tau}||g_\tau||_2^2} + \delta}{\alpha} \le \frac{G + \delta}{\alpha}
> \end{align}
>
> where $g_\tau$ is a stochastic gradient at time $\tau$ and $\beta$ is a momentum parameter for second-order momentum of Adam. Therefore, we believe that there is no problem. From the above equation, we can see that $\delta$ cannot make the upper bound of $1/\gamma$ arbitrarily small whatever the value of $\delta$ is due to the existence of $G$. Rather, as we note in our paper (see remark "advantages of using adaptive gradient methods in Theorem 1''), $1/\gamma$ could be smaller when the **gradients are sparse** (when $||g_\tau||_2$ is small, meaning the gradient is sparse) which coincides with the convex regret theory in [8, 9, 10].
>
> ---------------------------------------------------------------------------------------------------
>
> [Q3: On subdifferentials]
>
> Thank you for the correction. However, with slight modification of subdifferential, $\ell_0$ regularization is still **valid** as a motivating example in our ProxGen framework. Toward this, we introduce the following additional quantity, ``limiting subdifferential'' [2, 11, 12].
> \begin{align*}
>     \partial \varphi(\bar\theta) = \lbrace u \in \mathbb{R}^d : \exists \theta_k \overset{\varphi}{\rightarrow} \bar\theta, u_k \in \widehat\partial \varphi(\theta_k), u_k \rightarrow u \rbrace
> \end{align*}
> where $\widehat{\partial} \varphi(\bar\theta)$ is a Frechet subdifferential in Definition 1 in our paper. Also, $\theta \overset{\varphi}{\rightarrow} \bar\theta$ means $\theta \rightarrow \bar\theta$ and $\varphi(\theta) \rightarrow \varphi(\bar\theta)$. The limiting subdifferential of $\ell_0$-norm is now well-defined and the convergence with this limiting subdifferential is also guaranteed without compromising the whole analysis in Appendix. We will incorporate this in our manuscript.
>
> ---------------------------------------------------------------------------------------------------
>
> [Q4: On conditions (C-3) and (C-4)]
>
> Thank you for your suggestion on condition (C-3), but unfortunately, that would not be working in our proof because there are both linear and quadratic terms for $\lVert \theta_{t+1} - \theta_{t} \rVert_2$ in right-hand side of the second inequality at Appendix line 763. Subtracting the quadratic term $(L + \rho_0 G) \lVert \theta_{t+1} - \theta_t \rVert_2^2$ cannot remove the linear term $\lVert \rho_{t} m_{t-1} \rVert_{2} \lVert \theta_{t+1} - \theta_{t} \rVert_{2}$, hence our condition (C-3) is necessary.
>
> However, as we show in Figure 1-(a) for training ResNet-34 on CIFAR-10 dataset, the constants $D$ and $G$ (both $1 \sim 3$) in condition (C-3) are negligible in practice compared to the total parameter dimension $d$ about $10^7$. In addition, it can be seen in Figure 1-(b) that $\lambda_{\mathrm{min}}(\alpha_t (C_t + \delta I)^{-1})$ is increasing over time, so the minimum eigenvalue of $\alpha_t (C_t + \delta I)^{-1}$ for all $t$ is guaranteed to be lower-bounded empirically. Furthermore, the condition (C-4) holds in theory as we derive in Appendix G for various optimization algorithms. Therefore, we note that our condition (C-3) and (C-4) are very mild. In fact, the condition (C-3) is also generally assumed in several previous studies in non-convex optimization (see references [19, 20, 40, 41, 42] in our paper).
>
> ---------------------------------------------------------------------------------------------------
>
> [Reference]
>
> [1] On the Convergence of a Class of Adam-type Algorithms in Non-convex Optimization, Chen et al., ICLR 2019.
>
> [2] Non-asymptotic Analysis of Stochastic Methods for Non-smooth Non-convex Regularized Problems, Xu et al., NeurIPS 2019.
>
> [3] Adaptive Methods for Nonconvex Optimization, Zaheer et al., NeurIPS 2018.
>
> [4] Towards Better Generalization of Adaptive Gradient Methods, Zhou et al., NeurIPS 2020.
>
> [5] Adaptive Federated Optimization, Reddi et al., ICLR 2021.
>
> [6] Private Stochastic Non-Convex Optimization: Adaptive Algorithms and Tighter Generalization Bounds, Zhou et al., 2020.
>
> [7] On Stochastic Moving-Average Estimators for Non-Convex Optimization, Guo et al., 2021.
>
> [8] Adaptive Subgradient Methods for Online Learning and Stochastic Optimization, Duchi et al., JMLR 2011.
>
> [9] Adam: A Method for Stochastic Optimization, Kingma and Ba, ICLR 2015.
>
> [10] On the Convergence of Adam and Beyond, Reddi et al., ICLR 2018.
>
> [11] Structured Sparsity Inducing Adaptive Optimizers for Deep Learning, Deleu et al., 2021.
>
> [12] Stochastic optimization for DC functions and non-smooth non-convex regularizers with non-asymptotic convergence, Xu et al., ICML 2019.

---

> > ### Comment · Reviewer_3MNN · 2021-08-18
> > **Additional comment**
> >
> > Thanks for the detailed comment which resolves some of my concern. However, after reading the comment, I   still have a concern that the paper need some more improvement.  I attach additional comments below but would increase the score if my concerns are resolved.
> >
> > ### Q1 Q2
> >
> > Since the convergence is taken care of by the $\delta$ term and using large batch, it really does not matter what $C_t$ is.  I would not be surprised to see an $1/\epsilon^4$ complexity result. By choosing a stepsize independent of the adaptive metric (and total iteration), the whole analysis is very similar to the analysis of deterministic proximal gradient descent, It just needs a very large batchsize to ensure that the error of approximating gradients are bounded to get a good trade-off. This is okay.
> >
> > So far it is still unclear to me what a rule the adaptive metric is playing in the theory.
> > Starting from line 224, the author makes several remarks about the complexity rate and draw comparison with existing work. However, their complexity result crucially depends on the parameter $Q_i$, which are only given in the last page of the appendix. I would recommend explaining these important constants  Q_1 Q_2, Q_3 before drawing any specific conclusions.
> >
> > Moreover,  the author says that adaptive metric reflects the advantage of the method. Would it be more advantageous if we set $C_t=0$, then the $1/\gamma$ as well as $Q_1$ will be minimized?
> >
> >
> >
> > ### Q4,
> > This is not what I meant. After subtraction, you will have a concave quadratic function w.r.t. $\Vert\theta_t-\theta_{t+1}\Vert$, which can be upper bounded by a term irrelevant to $\Vert\theta_t-\theta_{t+1}\Vert$. This is a standard trick in stochastic optimization and may be helpful.
> >
> > The authors mentioned that C3 is standard. I checked the reference but still could not find it in the reference [40, 42]. Can you point out explicitly in the reference?
> >
> > The author mentioned about a constraint set $\Omega$ in (1), what is $\Omega$? And is it correct to have the optimality condition in line 756?

---

> > > ### Author Response · Authors · 2021-08-23
> > > **Response to Additional Comments**
> > >
> > > Thank you very much for insightful comments. We apologize for the slow response -- we wanted to do justice to your points, including the new proof direction you suggested.
> > >
> > > Please find our response below. Thank you for letting us know if you have any additional questions.
> > >
> > > -----------------------------------------
> > >
> > > [Q1: On theory.]
> > >
> > > Note that it is well-known that the first-order methods can achieve no better than $\mathcal{O}(1/\epsilon^4)$ convergence rate [A, B] that can be achieved by the existing theories for vanilla SGD as well as adaptive gradient methods under some settings. Our work is the first study achieving this optimal bound for a family of stochastic proximal gradient methods in the {\bf most general form}, incorporating the momentum $m_t$ , preconditioning matrix $C_t$, non-smooth and non-convex regularized problems.
> > >
> > > Focusing only on adaptivity, as the reviewer recognized, the adaptivity of the solver determines $\gamma$ assumed in (C-4) and $\gamma$ contributes to determining constants $Q_1 \sim Q_3$ of our theory. That is, our theory provides a tool to investigate how the adaptive gradient methods could be advantageous via $\gamma$.
> > > Here, the role of $\delta$ is controllable in our theory. Specifically, in our theory, $\delta$ is a free parameter, and the optimal bound $\mathcal{O}(1/\epsilon^4)$ can be obtained regardless of this value. When $\delta$ is very large, as the reviewer said, the role of $C_t$ becomes very small and the solver behaves similarly to vanilla SGD, but when $\delta$ is small, the adaptive characteristic according to $C_t$ (hence $\gamma$) really matters. We will clarify this in the revision.
> > >
> > > Regarding the constants $Q_1 \sim Q_3$, our analysis does not appeal to that the adaptive gradient methods are *always* better than vanilla SGD. Rather, we emphasize that the adaptive gradient methods could *potentially lead to huge improvements* in terms of convergence through $\gamma$ in our condition (C-4) in certain scenarios (this is exactly the same situation in AdaGrad, special case of our framework for the convex loss and regularizer). As a representative example, in line 236 in our paper and rebuttal, we show that the quantity $1/\gamma$ for Adam could be dramatically small when the gradients are sparse. As the reviewer suggested, we will explain the dependency of $\gamma$ on $Q_1 \sim Q_3$ more explicitly in the main text rather than in the appendix and incorporate our arguments above so that there is no misunderstanding. In fact, the constants $Q_1, Q_2,$ and $Q_3$ are affinely proportional to $1/\gamma$ as $a + b \times 1/\gamma$ for some constant $a$ and $b$ where $b$ is not smaller than $a$.
> > >
> > > -----------------------------------------
> > >
> > > [Q2: On case of $C_t = 0$.]
> > >
> > > As the reviewer said, in case of $C_t = 0$ and $\delta = 1$ (corresponding to vanilla SGD), the $1/\gamma$ is computed as $1/\alpha$ for constant stepsize (for easy comparison, we adopt the constant stepsize). As discussed in our paper at line 236 and our rebuttal, the $1/\gamma$ for Adam with constant stepsize is computed as
> > > \begin{align*}
> > >     Q_i \propto 1/\gamma = \frac{\sqrt{(1 - \beta) \sum\limits_{\tau=1}^{t}\beta^{t-\tau}\lVert g_\tau\rVert_2^2} + \delta}{\alpha}
> > > \end{align*}
> > > Here, if $\lVert g_\tau\rVert_2 < 1$ holds roughly where $\delta$ is a small constant here, we conclude that Adam can have lower value of $1/\gamma$ when the gradient is sparse, thereby achieving better convergence rate than vanilla SGD.
> > >
> > > Again, note that we are not claiming Adam is always better than vanilla SGD in our theory. Rather in certain scenarios Adam can achieve much smaller convergence bounds than vanilla SGD.
> > >
> > > -----------------------------------------
> > >
> > > [Q3: On condition (C-3).]
> > >
> > > In reference [40], the Assumption 2 in Section 4.1 (stagewise SGD) assumes the bounded domain which is stronger than our (C-3)-(i), and the Assumption 3 in Section 4.2 (stagewise AdaGrad) implies the bounded gradient which is weaker than our (C-3)-(ii). However, under the Assumption 1-(ii) in [40], we could conclude that the $\ell_2$-norm of the gradient is also bounded almost surely. In reference [42], the Assumption A2 in Section 3 is the exactly the same as our (C-3)-(ii). In Theorem 3.1 in [42], the authors further assume the quantity $\lVert \alpha_t m_t / \sqrt{\widehat v_t}\rVert_2$ is bounded where $\theta_{t+1} - \theta_t = - \alpha_t m_t / \sqrt{\widehat{v}_t}$, hence this condition says that the final step vector is finite, which is also the exactly the same as our (C-3)-(i).
> > >
> > > -----------------------------------------
> > >
> > > [Q4: On the proof.]
> > >
> > > Thank you for your valuable suggestion. As the reviewer said, after subtraction, we could bound the concave quadratic term in $\lVert \theta_{t+1} - \theta_t \rVert_2$ with some constant $M$. However, in line 773, we should take the summation over the timestamp $t = 0, 1, \cdots, T-1$, then the right-hand side of the inequality in line 733 will involve $M \times T$, which would be problematic. Therefore, it is non-trivial to remove the condition (C-3)-(i), but we plan to focus on it as future work.
> > >
> > > -----------------------------------------
> > >
> > > [Q5: On the last comment.]
> > >
> > > In order to allow general domain, we denote $\Omega$ but consider a real space $\Omega = \mathbb{R}^{p}$ in our paper where $p$ is the problem dimension. Regarding the optimality condition, in detail, we could obtain the line 756 and 757 by Exercise 8.8-(c) and Theorem 10.1 in [D]. We will clarify this point in the revision.
> > >
> > > -----------------------------------------
> > >
> > > [References]
> > >
> > > [A] Lower Bounds for Non-convex Stochastic Optimization, Arjevani et al., 2019.
> > >
> > > [B] Lower Bounds for Finding Stationary Points, Carmon et al., 2017.
> > >
> > > [C] Adaptive Subgradient Methods for Online Learning and Stochastic Optimization, Duchi et al., JMLR 2011.
> > >
> > > [D] Variational Analysis, Rockafellar, R. Tyrrell, Wets, Roger J-B, Springer, 1988.

---

> > > > ### Comment · Reviewer_3MNN · 2021-08-25
> > > > **Thanks for the reply**
> > > >
> > > > Thanks for the detailed comment. I will gladly raise the score to 5 since the author resolves some of my concern.  However, I am still concerned about the technical novelty and the soundness of theoretical contribution.
> > > >
> > > > [Q2: On case of Ct=0.]
> > > > Can we let $C_t=0$ and $\delta$ be small constant at the same time? That appears to give the optimal setting.
> > > >
> > > > Besides, I am still not convinced by the theoretical advantage presented in the main theorem. While Q_i depends on Ct and delta, it also depends on parameters L which implicitly depends on the dimension d. In contrast, AdaGrad has no dependence on L and it is very clear from their paper how the complexity depends on each parameter.
> > > >
> > > > Q5
> > > > The only nontrivial case of using $\Omega$ is when $\Omega$ is a closed bounded set, which automatically implies C-3-i). In this case, the optimality in line 756 needs to involve the subdifferential of indicator function on $\Omega$.
> > > > I am Okay with the optimality condition If $\Omega$ is the whole Euclidean space. However, it is very confusing to write $x\in \Omega$ when the problem is actually unconstrained.

---

> > > > > ### Author Response · Authors · 2021-08-29
> > > > > **Response to Additional Comments**
> > > > >
> > > > > Thank you for your additional comments.
> > > > >
> > > > > ----------------------------------------------
> > > > >
> > > > > [On the case of $C_t = O$.]
> > > > >
> > > > > We would like to point out that $C_t = 0$ and small $\delta$ are **not optimal** in terms of $1/\gamma$.
> > > > > When $C_t = 0$ (that is, $1/\gamma = \delta / \alpha$), as $\delta \rightarrow 0$, we can get $1/\gamma$ approaching to zero, but that will violate our stepsize condition $\alpha \le \delta / 3L$ in Theorem 1. Therefore, in our theory, we need to reduce $\delta$ and $\alpha$ in the **same** scale and the quantity $1/\gamma$ remains the same (the constants $Q_i$s will not change!) We believe that this will address reviewer's concern and it does not compromise the technical novelty or soundness of our theory.
> > > > >
> > > > > Also note that the dependency on the smoothness parameter $L$ comes from the consideration of non-convexity. This is the same for almost all studies dealing with non-convex optimization (The dependency on $L$ comes directly from the $L$-smoothness assumption (C-1), and as mentioned in the paper, this assumption is very standard for non-convex analysis). Even in the studies of extending AdaGrad to the non-convex setting, the same smoothness condition is assumed, and thus, implicit bounds are naturally obtained through the smoothness parameter $L$ [A, B].
> > > > >
> > > > > As such, $L$ smoothness is a very standard condition, and through this condition, a broader and more practical function can be considered (especially in the situation of considering a non-convex deep learning model). We are therefore convinced that this is not a drawback of our analysis.
> > > > >
> > > > > ----------------------------------------------
> > > > >
> > > > > [On domain $\Omega$.]
> > > > >
> > > > > Thank you for pointing this out. As the reviewer noted, for clarity, we will restrict the domain $\Omega = \mathbb{R}^{p}$. Although we set $\Omega = \mathbb{R}^p$, our condition (C-3)-(i) could handle the bounded domain cases to some extent.
> > > > >
> > > > > ----------------------------------------------
> > > > >
> > > > > [References]
> > > > >
> > > > > [A] On the Convergence of a Class of Adam-type Algorithms in Non-convex Optimization, Chen et al., ICLR 2019.
> > > > >
> > > > > [B] Stochastic Optimization for DC Functions and Non-smooth Non-convex Regularizers with Non-asymptotic Convergence, Xu et al., ICML 2019.

---

### Official Review · Reviewer_PpkK · 2021-07-29

**Rating:** 6
**Confidence:** 3

**Summary:**

This paper gives a proximal gradient method for optimizing nonsmooth and possibly nonconvex regularizers. It could be applied to impose structure on neural networks such as binary quantization.

**Limitations And Societal Impact:**

Some additional questions: Figure 2 shows the constant factors in the assumptions C1-C4, is it possible to compare with the convergence, i.e. |\nabla F| side by side to show whether the convergence theory is tight?

**Main Review:**

The papers reasonably propose proximal methods to handle these regularizers. As proximal mapping is more progressive than subgradient descent, so it should be straightforward.

One of my questions is about the technical difficulty in developing the theory of proof, as to determine whether it qualifies to be in Neurips. The paper is more of an experimental study overall.

**Time Spent Reviewing:**

1

---

> ### Author Response · Authors · 2021-08-10
> **Response to Reviewer PpkK**
>
> Dear Reviewer PpkK,
>
> Thank you for your constructive feedback.
>
> ---------------------------------------------------
>
> [Q1: On technical difficulties of theory]
>
> The main challenging issue, compared to the most relevant work [A], is **how to handle the first-order momentum $m_t$ and the arbitrary preconditioner $C_t$** as discussed in Section 3 of our paper (lines 192-209) "challenges specific to the analysis of ProxGen''. Since [A] only guarantees the convergence for the proximal gradient descent of vanilla SGD, their proof technique is not trivially applicable to our case. Introducing conditions (C-3) and (C-4), we were able to solve this challenge in a totally different way. Due to the existence of $m_t$, it is highly non-trivial to bound the term $\lVert m_t - \nabla f(\theta_t)\rVert_2$ whereas $\lVert g_t - \nabla f(\theta_t)\rVert_2$ in [A] can be easily bounded by (C-2). Also, we need to handle the quadratic term $(\theta - \theta_t)^\mathsf{T} (C_t + \delta I) (\theta - \theta_t)$ in Algorithm 1 which is no problem in [A] since $C_t$ is absent in [A]. Eventually, we can successfully guarantee the convergence rate $\mathcal{O}(1/\epsilon^4)$ under suitable batch size condition as in Corollary 1.
>
> ---------------------------------------------------
>
> [Q2: On Figure 2 in theory]
>
> Thank you for the interesting question. Unfortunately, it is not possible to explicitly calculate how tight the constants in our theory are in practice. This is because the neural network we are considering makes this calculation of the constants in (C-1) and (C-2) very complicated (please note that in Figure 1 we only computes the constants for (C-3) and (C-4)):
> computing the smoothness constant $L$ in (C-1) for very complex function is intractable. Also, in order to compute the parameter $\sigma$ in condition (C-2), one should consider the stochastic gradient evaluated at each data point, which is also intractable for large dataset usually considered in modern deep learning tasks.
>
> For this reason, among **all** the papers studying optimization of neural networks we know, including all the papers we have referenced, there is no single paper that explicitly obtains the constants in bounds and confirms their tightness.
>
> ---------------------------------------------------
>
> [References]
>
> [A] Non-asymptotic Analysis of Stochastic Methods for Non-smooth Non-convex Regularized Problems, Xu et al., NeurIPS 2019.

---

### Decision · Program_Chairs · 2021-09-27

**Decision:**

Accept (Poster)

**Comment:**

Reviews were initially highly polarized: while Uvqk and REN2 give a positive score and qualify this as "well-executed" and "authors generally do a good job in arguing the empirical and theoretical advantages of the proposed method", reviewer 3MNN argues that "the convergence analysis is not technically sound". The authors provided a strong rebuttal that increased the score of reviewer 3MNN.

After the discussion period, the reviewers agree that this is a contribution that can be of high interest to the NeurIPS community, and as such my recommendation is to accept for publication. However, reviewer 3MNN raises some valid points regarding comparison of constants with existing results and regarding clarity of the statements that I encourage the authors to incorporate into the final version.